# Distinct patterns of activity in individual cortical neurons and local networks in primary somatosensory cortex of mice evoked by square-wave mechanical limb stimulation

**Mischa V. Bandet** [1,2], **Bin Dong** [2,3], **Ian R. Winship** [1,2,3] *

1 Neuroscience and Mental Health Institute, University of Alberta, Edmonton, Alberta, Canada,
2 Neurochemical Research Unit, University of Alberta, Edmonton, Alberta, Canada, 3 Department of Psychiatry, University of Alberta, Edmonton, Alberta, Canada

* iwinship@ualberta.ca

**Data Availability Statement:** The data underlying the results presented in the study are available from https://osf.io/rhfqy/.

## Abstract

Artificial forms of mechanical limb stimulation are used within multiple fields of study to determine the level of cortical excitability and to map the trajectory of neuronal recovery from cortical damage or disease. Square-wave mechanical or electrical stimuli are often used in these studies, but a characterization of sensory-evoked response properties to square-waves with distinct fundamental frequencies but overlapping harmonics has not been performed. To distinguish between somatic stimuli, the primary somatosensory cortex must be able to represent distinct stimuli with unique patterns of activity, even if they have overlapping features. Thus, mechanical square-wave stimulation was used in conjunction with regional and cellular imaging to examine regional and cellular response properties evoked by different frequencies of stimulation. Flavoprotein autofluorescence imaging was used to map the somatosensory cortex of anaesthetized C57BL/6 mice, and *in vivo* two-photon $Ca^{2+}$ imaging was used to define patterns of neuronal activation during mechanical square-wave stimulation of the contralateral forelimb or hindlimb at various frequencies (3, 10, 100, 200, and 300 Hz). The data revealed that neurons within the limb associated somatosensory cortex responding to various frequencies of square-wave stimuli exhibit stimulus-specific patterns of activity. Subsets of neurons were found to have sensory-evoked activity that is either primarily responsive to single stimulus frequencies or broadly responsive to multiple frequencies of limb stimulation. High frequency stimuli were shown to elicit more population activity, with a greater percentage of the population responding and greater percentage of cells with high amplitude responses. Stimulus-evoked cell-cell correlations within these neuronal networks varied as a function of frequency of stimulation, such that each stimulus elicited a distinct pattern that was more consistent across multiple trials of the same stimulus compared to trials at different frequencies of stimulation. The variation in cortical response to different square-wave stimuli can thus be represented by the

**Funding:** IRW Alberta Innovates Health Solutions https://albertainnovates.ca Natural Sciences and Engineering Research Council of Canada https://www.nserc-crsng.gc.ca Canada Foundation for Innovation https://www.innovation.ca Province of Alberta Small Equipment Grants program MVB Canadian Institutes of Health Research https://cihr-irsc.gc.ca Natural Sciences and Engineering Research Council of Canada https://www.nserc-crsng.gc.ca Queen Elizabeth II Graduate Studentships.

**Competing interests:** The authors have declared that no competing interests exist.

population pattern of supra-threshold $Ca^{2+}$ transients, the magnitude and temporal properties of the evoked activity, and the structure of the stimulus-evoked correlation between neurons.

## Introduction

Investigations of how the somatosensory cortex responds to artificial forms of stimulation, and to what extent patterns of regional and cellular activity can distinguish between distinct stimuli with overlapping characteristics, is important in interpreting studies that use such stimuli to elicit cortical responses in the healthy brain, or as a measure of cortical excitability and plasticity during disease or after injury [1–5]. While square-wave and sinusoidal patterns of mechanical limb stimulation have been used in studies after cortical injury, and those studies identified deficits in the amplitude and fidelity of evoked activity after damage to the somatosensory cortex (for example, see [1,2,4,6]), a more detailed investigation across multiple frequencies and intensities of stimulation has not been performed with mechanical square-wave stimuli. Mechanical square-wave stimulation allows precise control of the fundamental frequency (as with sinusoidal stimulation) but includes harmonic frequencies. These harmonics may overlap between different stimuli, and it may, therefore, be more challenging for the somatosensory system to distinguish between these stimuli. Whereas considerable literature exists for somatosensation within the healthy brain in non-human primates (for review, see [7,8]), and the sensory-evoked response properties of the barrel cortex of rodents has been extensively studied [9,10], literature on the sensory evoked response properties of limb associated somatosensory system of rodents is more limited. Given that rodents are a prominent animal model used to study recovery of reaching and upper limb use after central nervous system damage or disability [11,12], it is important to understand how different forms of somatic and movement related stimuli used in rodent research are processed in the rodent somatosensory cortex. Most current research on the limb-associated somatosensory cortex in rodents utilize artificial forms of mechanical [1,13–17] or electrical stimuli [18–22] to assess sensory-evoked responses using electrophysiology of individual cells or aggregate responses from large cortical regions. Artificial stimuli used for rodent research that may be thought of as relatively simple in comparison to naturalistic stimuli or movements, such as limb oscillation through a single axis of motion, still likely result in the generation of complex multimodal sensory information in the periphery. These multimodal signals may arise from activation of mechanoreceptors at the stimulator attachment site, the propagation of vibration at the fundamental and harmonic frequencies to distal Pacinian corpuscles closely associated with joints and bones [23], the activation of proprioceptive receptors within the muscles and joints, and to a large potential array of signals arising due to deflections of hair follicle associated mechanoreceptors (for review, see [24]). Although particular subsets of cutaneous and proprioceptive mechanoreceptors have long been known in non-human primates to be tuned to characteristics such as stimulus intensity, frequency and receptive field location [7,8,25–28], the harmonics resulting from square-wave oscillation of the entire forelimb or hindlimb of mice would likely elicit highly mixed activity in mechanoreceptor populations, with weighted preference to mechanoreceptor populations most sensitive to the particular frequencies and biomechanics of the stimulus.

Calcium imaging has been used as a method for studying activity within regional cortical networks [1,29–32]. Here, $Ca^{2+}$ imaging was used to investigate how the limb-associated somatosensory cortex represents different frequencies of square-wave limb oscillation, and if differential patterns of activity in somatosensory neurons and local networks could be detected

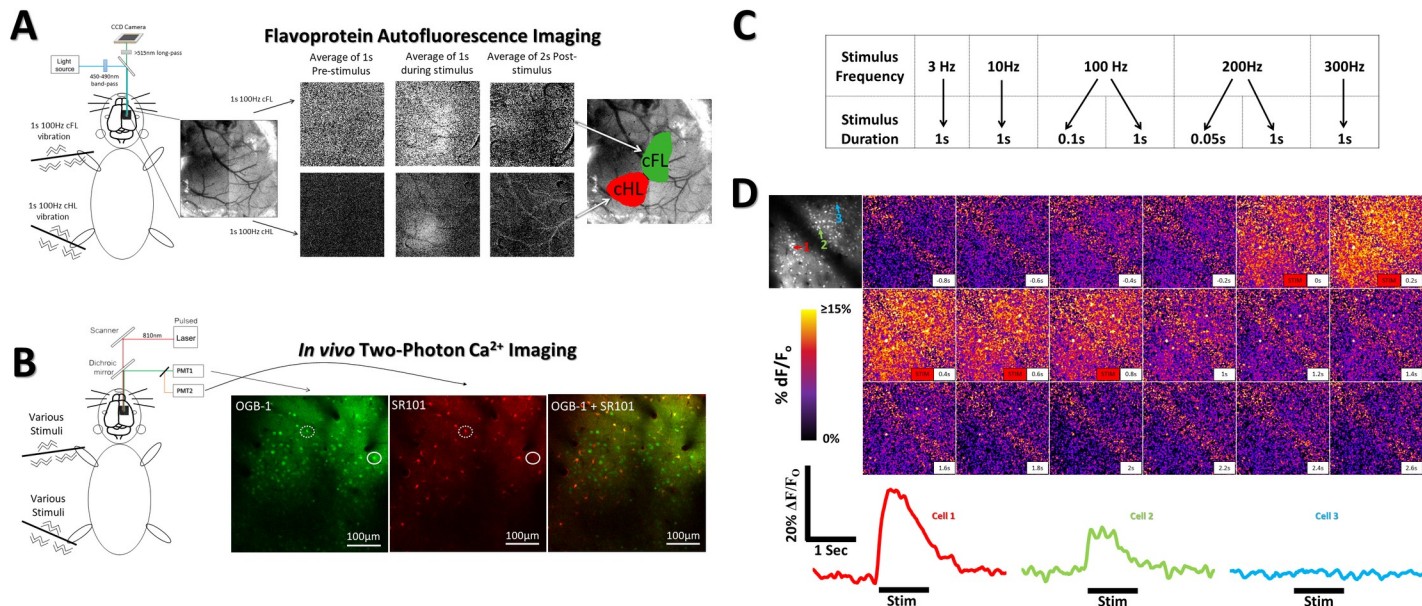

**Fig 1. Imaging protocols and experimental methods.** (*A*) Oscillatory stimulation (1s, 100 Hz) of the cFL or cHL during imaging of flavoprotein autofluorescence (FA) was used to define S1FL and S1HL limb associated somatosensory regions, respectively. The time course and regional distribution of FA response to cFL (top) and cHL (bottom) stimulation are shown. Each frame displays the averaged $\Delta F/F_o$ fluorescence intensity over the period indicated. FA response maps (see Materials and Methods) were thresholded at 50% of peak response amplitude and merged with an image of the surface vasculature to create color-coded regional maps of cFL and cHL activation. Regional maps from FA imaging were used as guides for membrane-permeant $Ca^{2+}$ indicator OGB-1 AM injection. Astrocyte marker SR101 was bath applied to delineate astrocytes from neurons. (*B*) Two-photon imaging setup showing OGB-1 and SR101 labelling in the S1HL of one animal. $Ca^{2+}$ imaging in both S1FL and S1HL during oscillatory stimulation was used to assess sensory-evoked single-cell responses. Circle depicts neuron not present on the SR101 astrocyte channel. Dotted circle depicts astrocyte labelling that was omitted from analysis. (*C*) 5 frequencies of stimulation were applied in random order for each animal. 8 trials of each stimulus were tested for each animal. (*D*) Cortical $Ca^{2+}$ response to a 1s 200Hz stimulation of the cHL of one example animal. Top left panel depicts an averaged image of the baseline period prior to stimulus onset within the cHL responsive region at 130um depth. Three example cells are indicated on the top left panel with their respective color coded $Ca^{2+}$ signals found below the $Ca^{2+}$ imaging montage.

within the $Ca^{2+}$ response. Flavoprotein autofluorescence imaging [33–37] was first used to identify limb-associated regions of cortical activation during mechanical forelimb or hindlimb stimulation delivered with piezoelectric actuators [1,38] (Fig 1A). *In vivo* two-photon $Ca^{2+}$ imaging was used to optically record the response properties of individual neurons and local neuronal networks within these limb-associated somatosensory regions during multiple frequencies of contralateral forelimb (cFL) or contralateral hindlimb (cHL) oscillation (Fig 1B–1D). Our data shows that the magnitude of the neuronal population response in somatosensory cortex is nonlinearly related to the frequency of mechanical limb stimulation. High frequency 100Hz, 200Hz and 300Hz stimuli were found to elicit more responding neurons, a greater response strength, and higher average cross-correlation between neurons than lower frequency 3Hz and 10Hz stimuli. Whereas average population response magnitude within the low frequency and high frequency groups was not a strong differentiator of how populations represented stimuli within these groups, the pattern of responsive neurons within the local neuronal network and differences observed in the cross-correlation maps for populations responding to these stimuli clearly differentiated between stimuli within and between stimulus groups.

## Materials and methods

### Animals

Male, two to four month old C57BL/6 mice (*n* = 10) were used in this study. Mice were group housed in standard laboratory cages in a temperature-controlled room (23°C), maintained on

a 12 hr light/dark cycle, and given standard laboratory diet and water *ad libitum*. All experiments were approved by the University of Alberta's Health Sciences Animal Care and Use Committee and adhered to the guidelines set by the Canadian Council for Animal Care. Animals were euthanized through decapitation following the end of the imaging session.

### Identification of forelimb and hindlimb somatosensory cortex representations

A surgical plane of anesthesia was achieved with 20% (w/v) urethane dissolved in saline and administered via intraperitoneal injection (1.25 g/kg; supplemented at 0.25 g/kg as needed). Body temperature was measured using a rectal probe and maintained at 37±0.5°C. Mice were administered 0.15 mL of Ringer's solution every 2 hours to maintain hydration levels. The skull was exposed by midline scalp incision and the skin retracted. A metal plate was secured to the skull using cyanoacrylate glue and dental cement and then fastened to the surgical stage to prevent head movement during imaging. A 4 x 4 mm region of the skull overlying the right hemisphere somatosensory region was thinned to 25–50% of original thickness using a high-speed dental drill (~1-5mm lateral, +2 to -2 mm posterior to bregma). This thinned region was covered with 1.3% low-melt agarose dissolved in artificial CSF (ACSF) at 37°C, then covered with a 3mm glass coverslip. Flavoprotein autofluorescence (FA) imaging was performed through this thin skull preparation before *in vivo* Ca$^{2+}$ imaging to determine somatosensory limb regions on the cortex. These mapped regions were later used for OGB-1 indicator injection and for determination of imaging window coordinates for Ca$^{2+}$ imaging following the FA mapping. For FA imaging, the cortical surface was illuminated with a xenon lamp (excitation band-pass filtered at 450-490nm). Autofluorescence emissions were long-pass filtered at 515nm and captured in 12-bit format by a Dalsa Pantera 1M60 camera mounted on a Leica SP5 confocal microscope. The depth of focus was set between 100–200 μm below the cortical surface.

Custom-made piezoceramic mechanical bending actuators were used to elicit oscillatory limb stimulation during FA and Ca$^{2+}$ imaging. The piezo bending actuators comprised a piezo element (Piezo Systems # Q220-A4-203YB) attached to an electrically insulated metal shaft holding a metal U-bend at its end. The palm of the mouse paw was placed within the U-bend, and the U-bend bent to shape to lightly secure the palm. The metal U-bend made contact across a vertical rectangular area of approximately 3x1mm on the palmar and dorsal surface of the hand. Stimulators were driven with square-wave signals from an A-M Systems Model 2100 Isolated Pulse Stimulator. For FA imaging only, stimulation alternated between contralateral forelimb (cFL) and contralateral hindlimb (cHL) for a total of 40 trials of stimulation of each limb. Placement of actuators was on the glabrous skin of the forepaw or hindpaw, with consistent alignment relative to the flexion of wrist and ankle. Images were captured for 5.0s at 5 Hz (1s before and 4s after stimulus onset; interstimulus interval = 20s). 40 trials for each limb were averaged in ImageJ software (NIH). Frames 1–1.5s after stimulus onset were averaged and divided by baseline frames 1s before stimulus onset to generate a response maps for each limb. Response maps were thresholded at 50% maximal response to determine limb associated response boundaries, merged, and overlaid on an image of surface vasculature to delineate the cFL and cHL somatosensory areas (Fig 1A). These areas were used as guides for Ca$^{2+}$ indicator injections [1], and were subsequently imaged.

### Calcium imaging

Subsequent to FA imaging as described above, the coverslip and agarose were removed and a 3 x 3 mm craniotomy performed centering over the cFL and cHL functional areas, determined

via FA imaging. A dental drill was used to progressively thin the overlying skull until the bone could be removed with forceps, leaving the dura intact. The exposed cortical surface was bathed in ACSF pre-heated to 37°C. Pressure injections of membrane-permeant Oregon Green BAPTA-1 (OGB-1) were made 180–200 µm below the cortical surface of the FL and HL cortical regions using glass micropipettes with resistances of 2–5 MΩ [1,32]. After OGB-1 injection, the cortex was incubated for 10min with sulforhodamine 101 (SR101) dissolved in DMSO to label astrocytes [1,32,38,39]. The craniotomy was then covered with 1.3% agarose dissolved in ACSF and sealed with a glass coverslip.

Two-photon imaging was performed using a Leica SP5 MP confocal microscope equipped with a Ti:Sapphire Coherent Chameleon Vision II laser tuned to 810 nm for OGB-1 and SR101 excitation. A Leica HCX PL APO L 20x 1.0NA water immersion objective was used. Images were acquired using Leica LAS AF using two line-averages, a zoom of 1.7x and a frame-rate of 25Hz. Images were acquired at 256x256 pixels over an area of 434x434µm, yielding a resolution of 1.7µm per pixel. $Ca^{2+}$ fluctuations in neurons, astrocytes, and neuropil were imaged in response to 8 trials at each different frequency and duration of mechanical stimulation (3, 10, 100, 200, and 300Hz for 1s, 0.1s 100Hz & 0.05s 200Hz). While imaging the cHL somatosensory region mapped via FA imaging, only the cHL was stimulated. Likewise, only the cFL was stimulated during imaging of the cFL somatosensory region. The same custom piezo-electric stimulators used for FA imaging were used for calcium imaging. The order of the stimulation frequency was randomized at the beginning of each experiment for each animal. During stimulation, the entire limb underwent an oscillation with the following peak-peak amplitudes by frequency based on the limb weight loaded electromechanical properties of the bending actuator: 280um (3 & 10Hz), 335um (100Hz), 220um (200Hz), 170um (300Hz). To determine these peak-peak oscillation amplitudes, the movement of the limb was imaged using a Dalsa Pantera 1M60 camera mounted on a Leica SP5 microscope using a 2.5X objective. The limb was attached to the stimulator and placed on a black background to increase contrast of the limb relative to the background. Bending actuators were driven by an A-M Systems Model 2100 stimulator, an analog stimulator with 1 microsecond timing (250 ns jitter). Three different imaging frame rates and resolutions were recorded for each frequency of limb stimulus based on the limits imposed by the camera hardware: 58fps at 1024x1024 (17.24ms exposure), 100fps at 512x512 (10ms exposure) and 157fps at 256x256 (6.37ms exposure). As all of these imaging frame rates were undersampling relative to the Nyquist criterion for the 300Hz stimuli, we employed several measures to get an index of the peak-to-peak amplitude of the limb oscillation as follows. For all imaging framerates, we recorded extended video (approximately 5 minutes per recording), then used max intensity projections to determine the maximal bending amplitude of the stimulus at each stimulus frequency. To confirm this method, we compared the max projection measurements and frame-per-frame measurements for low frequency stimuli and found them to be equal. Peak-to-peak deviation measurements from all three imaging framerates at all frequencies were compared and found not to be significantly different. To determine if the peak-to-peak bending amplitude was consistent across multiple stimulus trials, we repeated the video recordings multiple times and separately analyzed each trial. Bending amplitudes were consistent across trials of the same stimulus frequency. Fig 1C depicts the various oscillatory stimuli used throughout this study to elicit $Ca^{2+}$ responses in cell populations. Fig 1D depicts an example timelapse montage of $Ca^{2+}$ imaging during limb stimulation and example cell responses.

## Two-photon image processing and determination of responding neurons

Using custom scripts written in Metamorph (Molecular Devices, California U.S.A.), a median filter (radius, 1 pixel) was applied to each of the image sequences of 8 trial sweeps at each

frequency and duration of stimulation in order to remove photodetector related photon transfer noise (e.g. photon-shot noise) from the digitized images. The 8 sweeps at each stimulus frequency and duration were independently averaged. Regions of interest (ROIs) were drawn around visible neurons and astrocytes were excluded from analysis by removing ROIs that co-labelled astrocytes in the SR101 channel. Raw neuronal fluorescence traces were exported from Metamorph and were imported into Excel (Microsoft, Inc.). A 5-point moving triangular filter was applied to each neuronal trace to remove noise artifacts in the temporal domain. We found that this triangular filter did not affect the overall amplitude or main signal properties of our recorded $Ca^{2+}$ traces, but did reduce single frame $Ca^{2+}$ trace artifacts. $\Delta F/F_o$ traces were generated from the raw fluorescence traces as previously described [1]. Neuronal and neuropil $Ca^{2+}$ signals were analyzed in Clampfit 10.0. Due to the large number of signal traces that required analysis (over 40,000 traces; over 6000 neurons; 10 mice), a range of threshold criteria, based on previous research [1,31,40–42], were tested to differentiate responsive neuron $Ca^{2+}$ transients from fluorescent noise. This range of threshold criteria was tested against manual annotation of $Ca^{2+}$ responses gathered from multiple observers for a small subset of the experimental dataset from multiple animals. Positive identification of a $Ca^{2+}$ transient from noise within the manual annotation was based on the expected $Ca^{2+}$ transient waveform demonstrating fast rise on the leading edge of the fluorescence, and a slow decay back to baseline. The threshold criteria found to most effectively select $Ca^{2+}$ transient waveforms that met this expected waveform shape was selected for identification of $Ca^{2+}$ transients across all animals. A threshold criteria requiring the $Ca^{2+}$ fluorescence of the cell, averaged from 8 trials, to increase by 3X the standard deviation of the baseline period $\Delta F/F_o$ (baseline defined as 1s before stimulus onset), and remain above this criteria for 160ms (4 successive frames), was used to differentiate the $Ca^{2+}$ transient waveform of a response of the neuron from random noise fluctuations. The percentage of neurons that met an additional criteria of peak $\Delta F/F_o$ greater than 10% were deemed "strongly responsive" neurons (Fig 3D) based on previous studies associating >10% increases in OGB-1 $\Delta F/F_o$ with firing of multiple action potentials underlying the $Ca^{2+}$ response [29]. Neuronal traces that met these criteria were included in subsequent analysis of peak amplitude and area under the curve (AUC) measurements. AUC was measured as the total area under the curve of the $\Delta F/F_o$ from when the stimulus began to when the $\Delta F/F_o$ returned to the baseline level. AUC was used as a measurement of the sum strength of the calcium transient response over the time course of that particular transient. Neuronal response maps for visualization (Fig 2A) were generated by averaging the calcium imaging frames during the 1s stimulus period for each stimuli and dividing by the average of the 1s baseline frames prior to stimulus onset. Percent of overlapping responsive neurons for each pairwise stimulus comparison (Fig 2G) was calculated by counting the number of cells showing common responses between the pairwise stimuli and dividing by the combined total population responsive to the pairwise stimuli. Percent overlapping responsive neurons was averaged across all animals (N = 10).

## Stimulus-evoked cell-cell correlation analysis

In order to demonstrate a visual example of how the calcium fluorescence of a single cell may correlate with other pixels across an imaging window, a custom Matlab R2018a (Mathworks) script was used to generate an example seed-based stimulus-evoked neuronal correlation map (as shown in Fig 4A). With this script, filtered image sequences of $\Delta F/F_o$ were imported into Matlab and the signal trace for a chosen neuron was correlated with the $\Delta F/Fo$ of all pixels within the imaging field. A representative image during 1s 100Hz stimulus measured at a depth of 130μm in the cHL region is shown in Fig 4A. Using a separate custom written script in Matlab, Pearson product-moment correlation coefficients were calculated between the z-

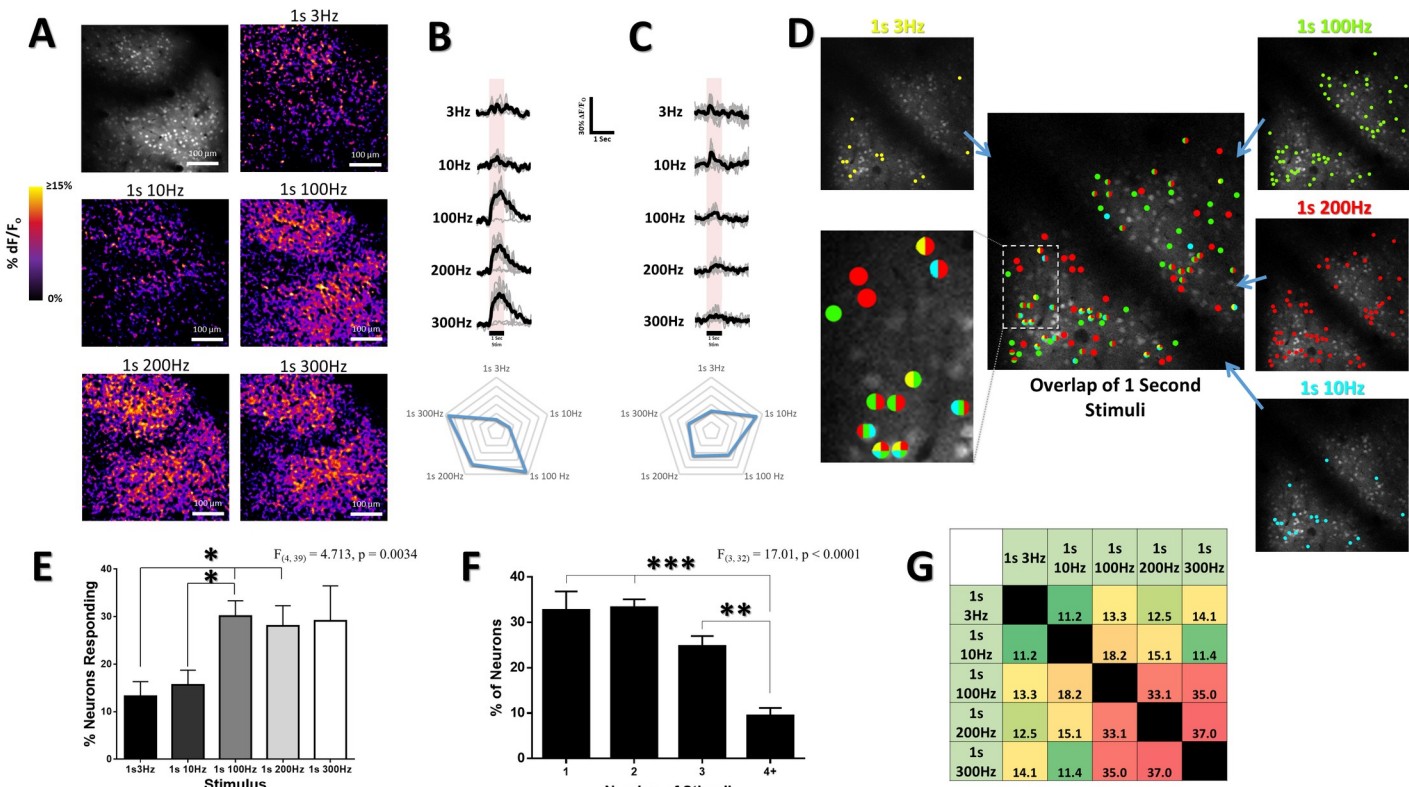

**Fig 2. Neuronal and population response selectivity to stimulus frequency within the limb associated somatosensory cortex.** (*A*) Difference images illustrating the averaged change in fluorescence $\Delta F/F_o$ from baseline over the 1s stimulus period. Top left panel depicts an averaged image of the baseline period prior to stimulus onset within the cHL responsive region at 130um depth in an example animal. High frequency 100, 200, & 300Hz stimuli elicit greater cortical activation of neurons and neuropil than lower frequency 3 & 10Hz stimuli (quantified in Fig 3). (*B,C*) $Ca^{2+}$ traces for individual neurons responding to the different stimuli (average of 8 trials of stimulation per limb) are shown for each stimulus for the particular chosen neuron. The neuron in *B* is more responsive to high frequency 100, 200, & 300Hz stimulation, and displays minimal response to lower frequency 3 and 10Hz stimuli. The neuron in *C* responds preferentially to 10Hz stimuli. Plots below $Ca^{2+}$ traces indicate peak $Ca^{2+}$ response magnitude for each stimulus (*D*) Color coded map of responsive neurons superimposed on an optical section at 130μm depth in the cHL of an example animal. Neurons were deemed responsive by threshold analysis of somatic responses, with color representing above-threshold responses to stimulation at the particular frequency and duration stated. High frequency stimuli (100 & 200Hz) elicit a greater number of neurons responsive for these stimuli. The inset image displays the same color code used for each particular stimulus and shows how certain cells within this population are selective in their response to particular stimuli, whereas others display overlap in their responsiveness to multiple different stimuli. (*E*) Mean percent of neurons responding to each of our stimuli across all animals (N = 10). (*F*) Mean percent of neurons within each optical section responsive to 1, 2, 3, or 4+ stimuli across all animals (N = 10). (*G*) Color coded chart displaying the percentage overlap between the neurons responsive to each stimulus for all animals (N = 10). High frequency 100, 200, & 300Hz stimuli display greater overlap in their responsive cell populations as compared to the activated cell population overlap for 3Hz and 10Hz stimuli. *p < 0.05; **p < 0.01; ***p < 0.001.

scored signal traces derived from each of our neuronal ROIs in a given optical section. Pairwise Pearson's r correlation coefficients were calculated between each pair of neurons in each optical section. These stimulus-evoked correlation coefficient maps (e.g. Figs 4B, 5 and 6) were created by plotting example matrices of these pairwise correlation coefficients at a specific region of S1 (FL or HL) during sensory stimulation of the limb. ROI location co-ordinates were exported from ROI in Metamorph and imported into Matlab in order to compute the pairwise distance between each ROI. The stimulus-evoked pairwise correlation coefficient for each neuron pair was measured as a function of the distance between the pair (for each frequency of stimulation), and a line fit for regression analysis. Determination of stimulus-evoked correlation map equality was performed using the Jennrich test for correlation matrix equality [43] implemented in Matlab (with p < 0.001 considered as significant). A stationary bootstrapping procedure was further used as a further test of the significance of the Pearson's r calculated between the $\Delta F/F_o$ of each neuron pair. Within the stationary bootstrapping procedure, 10000

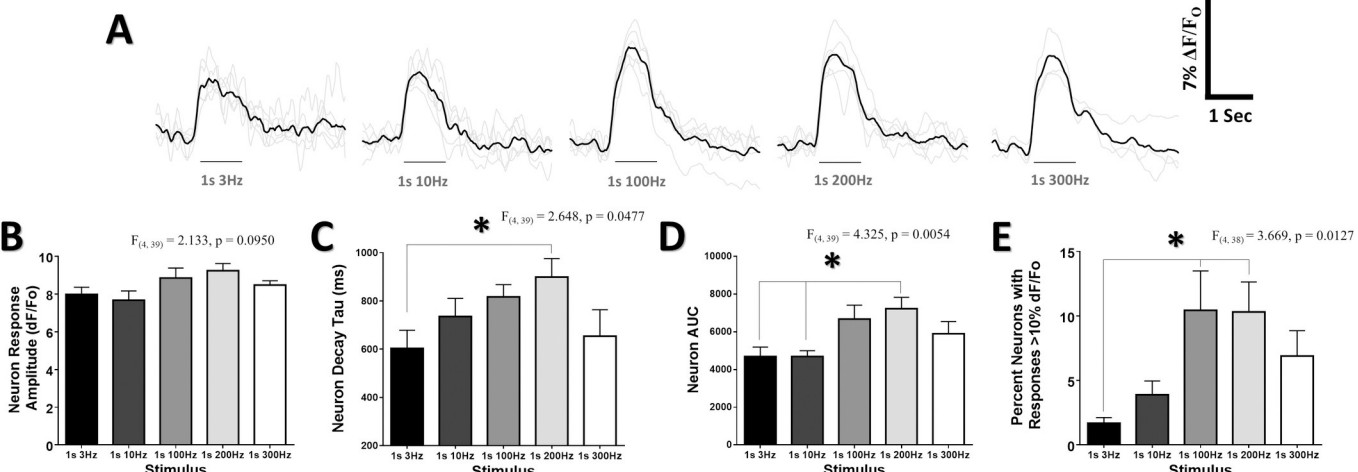

**Fig 3. Mean population Ca2+ response characteristics do not scale linearly with the number of oscillations in a stimulus.** (*A*) $Ca^{2+}$ response traces averaged from all cells at 130μm in the cHL of all animals for each frequency. Grey lines depict individual animal $Ca^{2+}$ responses, black lines are the average of the grey lines from all animals. Average data for b,c,d, & e are taken from all limb representations from all animals. (*B*) No change in the mean neuron $Ca^{2+}$ response amplitude was observed for any of the stimulus frequencies. A significant effect of frequency was observed for the under the curve (AUC) (*C*) of neuron responses to the different frequencies of stimulation. (*D*) A significant effect of frequency was observed on the percentage of neurons with responses greater than 10% $\Delta F/F_o$. *$p < 0.05$; **$p < 0.01$; ***$p < 0.001$.

iterations were run and a mean block length of 5 frames (200ms) was used based on previous studies indicating the minimum period of time between action potentials necessary to detect a calcium transient using OGB-1 [44]. Pairwise r values greater than the 97.5th percentile of the stationary bootstrap are indicated by light blue pixels (Fig 4C) and the correlation map replotted to show only those correlations meeting the bootstrap criteria (Fig 4D). To further determine the degree of correlation map similarity, comparisons of correlation map similarity within repeated trials of the same stimulus, and between trials of different stimuli, were generated using custom Matlab based scripts based on the structural similarity index (SSIM). The SSIM score assessed the localized correlation value, contrast, and structure of correlation maps it was comparing. The SSIM score has a maximal value of 1 that indicates complete similarity between the two maps in comparison. An example of these within and between stimulus comparisons is shown in S1 Fig. SSIM score results were averaged across all animals (N = 10).

## Statistical analysis

Univariate comparisons were made using ANOVA with *post hoc* Tukey's honestly significant difference (HSD) tests for all statistical tests. Stimulus-evoked correlation between cells was analyzed using the Pearson *r*. Significance for the slope of the pairwise neuron correlation coefficient by distance was analyzed using multiple t-tests between the mean slope value for each stimulus condition and a no-slope null hypothesis, and adjusted using Bonferroni for multiple comparisons. Statistical analyses were performed in GraphPad Prism 6.0 windows version. A *p* value of ≤0.05 was considered statistically significant. A *p* value of ≤0.10 was considered a trend, however not statistically significant. Data are expressed as the mean ± SEM.

## Results

### Stimulus frequency preference within somatosensory cortex neurons

To elucidate the effect of the frequency of oscillatory limb stimulation on $Ca^{2+}$ response strength at the level of individual neurons within the population, $Ca^{2+}$ transients evoked by

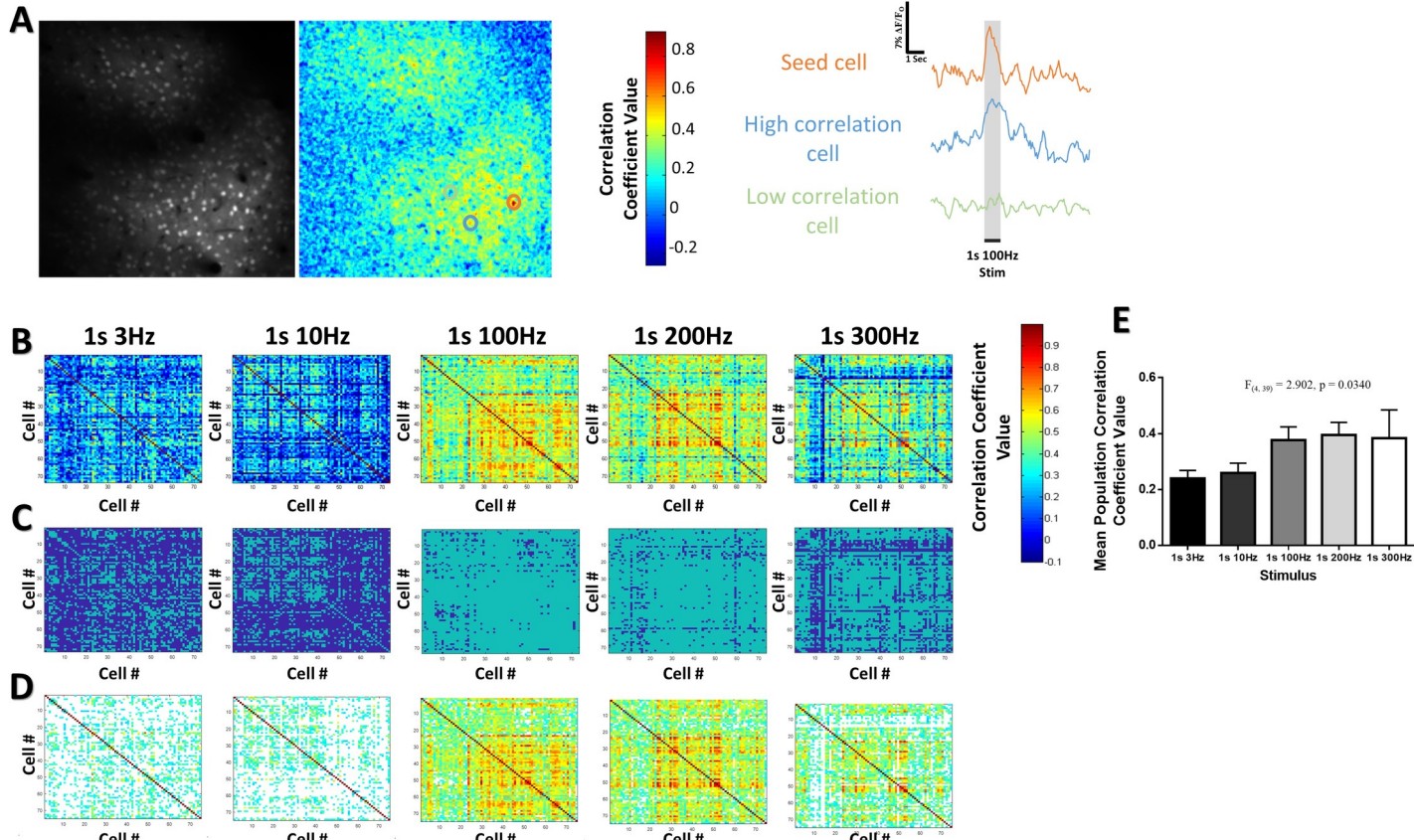

**Fig 4. Stimulus-evoked pairwise correlations vary as a function of the stimulus frequency.** (*A*) Example of a seed-cell based correlation technique for a 1s 100Hz stimulus at 130μm in the cHL of one animal. The $\Delta F/F_o$ fluorescence over time for the seed cell (orange) was correlated against the fluorescence over time for all other pixels in the imaging frame to generate the correlation map displayed. Example fluorescence traces are shown to the right with an example of a cell with high (blue trace) and low (green trace) correlation to the seed cell. (*B*) Example heat map matrices of the stimulus-evoked correlation coefficient between a population of 74 neurons within the cFL somatosensory cortex of one example animal. (*C*) Bootstrapping was used to determine pairwise correlations greater than the 97.5[th] percentile of the bootstrap and indicated by light blue (see methods). (*D*) The pairwise correlation values were re-plotted with the correlations that did not meet the bootstrap criteria excluded. (*E*) A significant effect of stimulus frequency on the mean population correlation coefficient value is observed across all animals (N = 10). $^*p < 0.05$; $^{**}p < 0.01$; $^{***}p < 0.001$.

square-wave mechanical limb oscillation were optically recorded during *in vivo* two-photon imaging. Fig 2A shows representative "response maps" from one animal illustrating post-stimulus increases in OGB-1 fluorescence in response to different frequencies of stimulation applied to the cHL for 1s each. In all animals imaged, differences in the $Ca^{2+}$ response amplitude and pattern of cortical activation for lower frequency 3/10 Hz stimuli in comparison to higher frequency 100/200/300 Hz stimuli are noticeable, with higher frequency stimuli appearing to result in a greater $Ca^{2+}$ response (quantified for all animals in Figs 2 and 3). Fig 2B and 2C depict representative neurons selected from the imaging plane shown in A that display different magnitudes in their $Ca^{2+}$ response to the different stimuli. Notably, the neuron in Fig 2B is more responsive to high frequency 100, 200, and 300Hz stimuli, whereas the neuron in Fig 2C exhibits a mean response that is transient and most strongly activated by 10Hz stimulation.

## Patterns of activation to somatosensory stimuli within S1 cortex are distinct

To evaluate the selectivity of neurons to particular square-wave stimuli across local networks of 100–350 cells in somatosensory cortex, threshold criteria were used to define cells

exhibiting a significant response to a particular stimulus. Fig 2D depicts a representative color-coded image plane of neurons exhibiting a significant (above threshold) response to each stimulus frequency. Variations in stimulus selectivity are apparent. Neurons selective to a single stimulus frequency are illustrated as a uniformly coloured dot, while neurons responding broadly to multiple frequencies are depicted with 2 or more colors. While individual neurons could be preferentially responsive to particular frequencies of stimulation, a significant main effect of stimulus frequency on the percent of neurons with above threshold responses was observed ($F_{(4, 39)}$ = 4.713, p = 0.0034) (Fig 2E). Notably, high frequency 100 & 200Hz stimuli displayed a significantly greater percentage of above threshold responses relative to 3 & 10Hz stimuli. Consistent with previous research by Hayashi et al., 2018, approximately 15% of neurons exhibited a significant response for the 3Hz or 10Hz stimuli, and approximately 30% of the population exhibited a significant response for each of the 100, 200, or 300Hz stimuli (Fig 2E). The majority of neurons with above threshold $Ca^{2+}$ responses were responding to multiple stimuli, with approximately 32.7 ± 4.2% selective to a single stimulus, 33.2 ± 1.8% activated by two stimuli, and 34.1 ± 3.8% activated by three or more stimuli (Fig 2F). 60.9% of all neurons that were responsive to 3 or more stimuli were responding to 100, 200, and 300Hz. To further determine the amount of overlap between above threshold responsive neurons for each stimulus, we measured the percentage of cells that demonstrated above threshold responses in common with other stimuli. Consistent with the large population of neurons showing preferential responses to high frequency 100, 200, & 300Hz stimuli, Fig 2G demonstrates that higher frequency stimuli displayed a larger percentage overlap in their responsive cell populations as compared to the percent overlap between lower frequency stimuli.

## Mean population $Ca^{2+}$ response characteristics do not scale linearly with the number of oscillations in a stimulus

Sensory-evoked $Ca^{2+}$ transients from neurons in S1 were analyzed to determine whether their response characteristics varied proportionately to the square-wave stimulus frequency. If response intensity were defined by the number of repetitive limb oscillations in a stimulus, it would be predicted that the amplitude and decay of the $Ca^{2+}$ transients would increase with each greater frequency. Fig 3A depicts mean $Ca^{2+}$ response traces from all above threshold cHL-responsive cells acquired at a fixed depth of 130um from the cortical surface in the S1HL. A greater overall response for high frequency 100, 200, and 300Hz stimuli relative to 3 and 10Hz stimuli is apparent. Quantification of $Ca^{2+}$ transients from all recorded limb responsive regions across all animals is shown in Fig 3B–3D. Notably, Fig 3B shows that the peak amplitude of the sensory evoked above-threshold $Ca^{2+}$ transient did not vary significantly according to stimulation frequency. However, a significant main effect of frequency was observed on AUC ($F_{(4, 39)}$ = 4.325, p = 0.0054), with 200Hz stimuli exhibiting significantly greater AUC than 3 & 10Hz stimuli (Fig 3C). No evidence of direct scaling of response properties, such that 10Hz would elicit greater neuron activity than 3Hz, or 200Hz greater activity than 100Hz, was observed.

Because a $\Delta F/F_o$ of greater than 10% (measured using OGB-1) has been associated with firing of multiple action potentials underlying the $Ca^{2+}$ response [29], the proportion of "strongly responsive" neurons with sensory-evoked $Ca^{2+}$ transient amplitude of greater than 10% $\Delta F/F_o$ was determined. The proportion of strongly responsive neurons varied with stimulation frequency ($F_{(4, 38)}$ = 3.669, p = 0.0127) (Fig 3D). Post hoc comparisons confirmed that a larger percentage of neurons exhibited peak amplitude of greater than 10% $\Delta F/F_o$ for 100 & 200Hz stimuli compared to 3Hz stimuli.

## Stimulus-evoked pairwise correlations vary as a function of the square-wave stimulus frequency

Analysis of the stimulus-evoked correlated activity between neurons was used to further define the structure of pairwise interrelationships of $Ca^{2+}$ fluctuations resulting from the sensory stimulus across populations of up to 350 neurons per optical section. These correlations provide an approximation of neurons that respond similarly, and thus allowed us to examine patterns of cortical neuron activity during different stimulus conditions. Fig 4A demonstrates a representative example of a seed-cell based stimulus-evoked correlation map for a 1s 100Hz stimulus where the fluorescence over time of the seed cell (orange) was correlated with fluorescence of all other pixels in the image. Neurons with highly correlated and weakly correlated stimulus-evoked activity are observed. Pairwise stimulus-evoked correlation strength between all neurons was also determined. Data for the stimulus-evoked correlation between all cells within network populations was pooled across animals and sorted by stimulus frequency. The distance between neuron pairs was determined by the Euclidean distance between the centers of mass of their cell bodies within the imaging plane. The stimulus-evoked correlation coefficient decreased as a function of the pairwise distance between neurons for all frequencies tested (mean slope of -0.26), a result predicted based on previous studies within the barrel cortex of mice that demonstrate higher correlations among nearby cells [45,46]. The slope of the line fit for the pairwise correlation as a function of distance was not significantly different between stimuli conditions ($F_{(4, 39)}$ = 0.4616, p = 0.7634), thereby indicating a similar decrease in the stimulus-evoked correlation by distance for all stimuli. Notably, all stimulus conditions exhibited a significant negative slope (Single sample t-test relative to theoretically slope of 0, *p* < .05 with Bonferroni correction for all stimulus conditions). Although this result may suggest a small degree of similarity in the stimulus related $Ca^{2+}$ measurements for neurons found closer together, the correlation between any two particular neurons is poorly predicted by the distance between them as indicated by very low $R^2$ values for all stimuli (mean $R^2$ of 0.029). The $R^2$ values for all stimuli were not significantly different from each other ($F_{(4, 39)}$ = 0.3522, p = 0.8409), suggesting that this low degree of predictive value did not vary based on the particular stimulus delivered. Thus, these data do not support spatial clustering in response to distinct stimuli. Fig 4B gives a color-coded depiction of the stimulus-evoked pairwise correlation between a population of 74 neurons within the cFL somatosensory cortex of a representative animal. Notably, the pattern of pairwise stimulus-evoked correlations varies between stimuli (*p* < 0.001, using Jennrich test of matrix equality [43]), thereby illustrating distinct inter-relationships of network activity resulting from different frequencies of stimulation. We further confirmed the significance of these pairwise cross correlations by running a stationary bootstrapping procedure (see methods). Bootstrapping further supported that the pairwise comparisons meeting the bootstrapping criteria varied as a function of the stimulus (Fig 4C and 4D). Across all animals, mean stimulus-evoked pairwise correlation varied significantly as a function of stimulation frequency ($F_{(4, 39)}$ = 2.902, p = 0.0340) (Fig 4E), with a trend towards greater overall stimulus-evoked correlation for high frequency stimuli.

## Frequency or quantity representations in cortical $Ca^{2+}$ transients

Stronger neuronal activation for high frequency 100, 200, & 300Hz stimuli, compared to 3 & 10Hz stimuli, is shown in previous data. However, because the frequency of oscillatory stimulation was varied but duration of stimulation was constant at 1s, the quantity of limb oscillations varied significantly between stimulus conditions. To clarify if stimulus frequency or absolute number of oscillations was the primary driver of distinct patterns of cortical activity, stimuli with 10 absolute oscillations of the limb during the stimulus period were delivered at

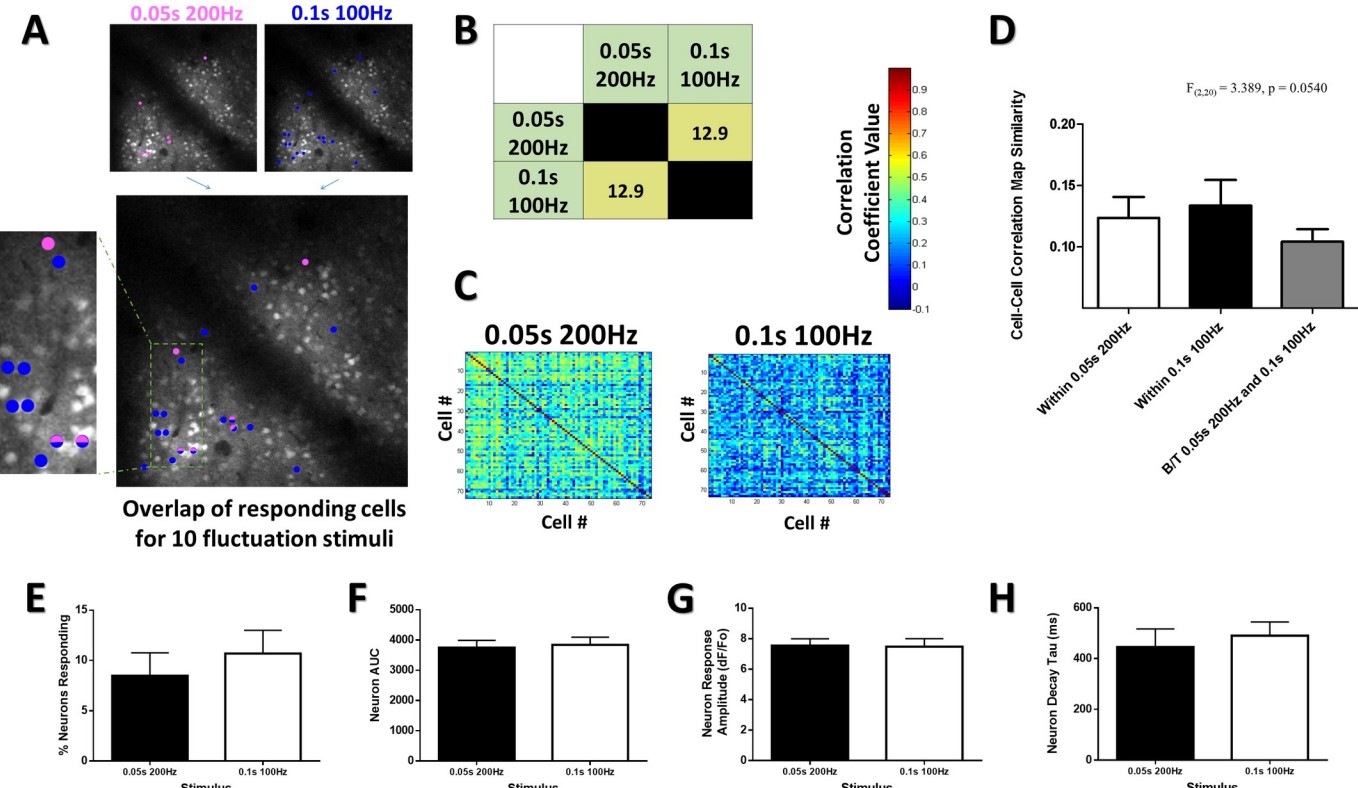

**Fig 5. Frequency or quantity representations in cortical Ca2+ transients.** (*A*) Color coded responsive neuron maps superimposed on an optical section at 130μm depth in the cHL of an example animal (same animal as in Fig 4). Neurons were deemed responsive by threshold analysis of somatic responses (see methods), with color representing above-threshold responses to stimulation at the particular frequency and duration stated. The overlapping image displays the same color code used for each particular stimulus and shows how certain cells within this population are selective in their response to particular stimuli, whereas others display overlap in their responsiveness to multiple different stimuli, despite all stimuli having the same number of limb oscillations (10 mechanical oscillations of the limb). (*B*) Color coded quantification chart displaying the percentage overlap between the neurons responsive to each stimulus quantified for all animals (N = 10). (*C*) Example heat map matrices of the pairwise stimulus-evoked correlation coefficient between a population of 74 neurons within the cFL somatosensory cortex of one example animal at 160μm depth for our 10 oscillation stimuli. (*D*) Structural similarity comparison of the maps for all trials depicts a trend towards greater map similarity for within stimulus conditions than between stimulus comparisons. No effect of frequency was seen for the percent of neurons responding (*E*), AUC (*F*), neuron response amplitude (*G*), or decay tau (*H*) for responsive neuron populations. *p < 0.05; **p < 0.01; ***p < 0.001.

two frequencies (0.05s at 200Hz and 0.1s at 100Hz). Selective patterns of evoked neuronal Ca²⁺ responses were still observed with these 10 oscillation stimuli (Fig 5A), with minimal overlap between neurons responsive to each stimulus (Fig 5B). Notably, when stimuli were restricted to 10 oscillations for these 0.05s at 200Hz and 0.1s at 100Hz trials, the lower percentage of responding neurons show reduced overlap relative to longer 1s at 200Hz and 1s at 100Hz stimuli (Figs 2G vs 5B). Fig 5C illustrates stimulus-evoked correlation maps between a population of 74 neurons within the cFL somatosensory cortex of a representative animal for these 10 oscillation stimuli (from the same animal as displayed in Fig 4B). To determine whether the stimulus-evoked correlations observed for these stimuli were repeated over multiple trails during the length of the imaging session, custom Matlab based routines were used to determine the similarity of these maps across trials to generate "within stimulus" comparisons. Stimulus-evoked correlation maps generated from one stimulus were also compared on a trial by trail basis to those generated from the other stimulus to generate "between stimulus" comparisons (see S1 Fig for SSIM method). We observed a statistical trend (p = 0.0540) suggesting higher stimulus-evoked correlation map similarity for within stimulus comparisons as opposed to comparisons between the two different 10 oscillation stimuli (Fig 5D). Observing data from all

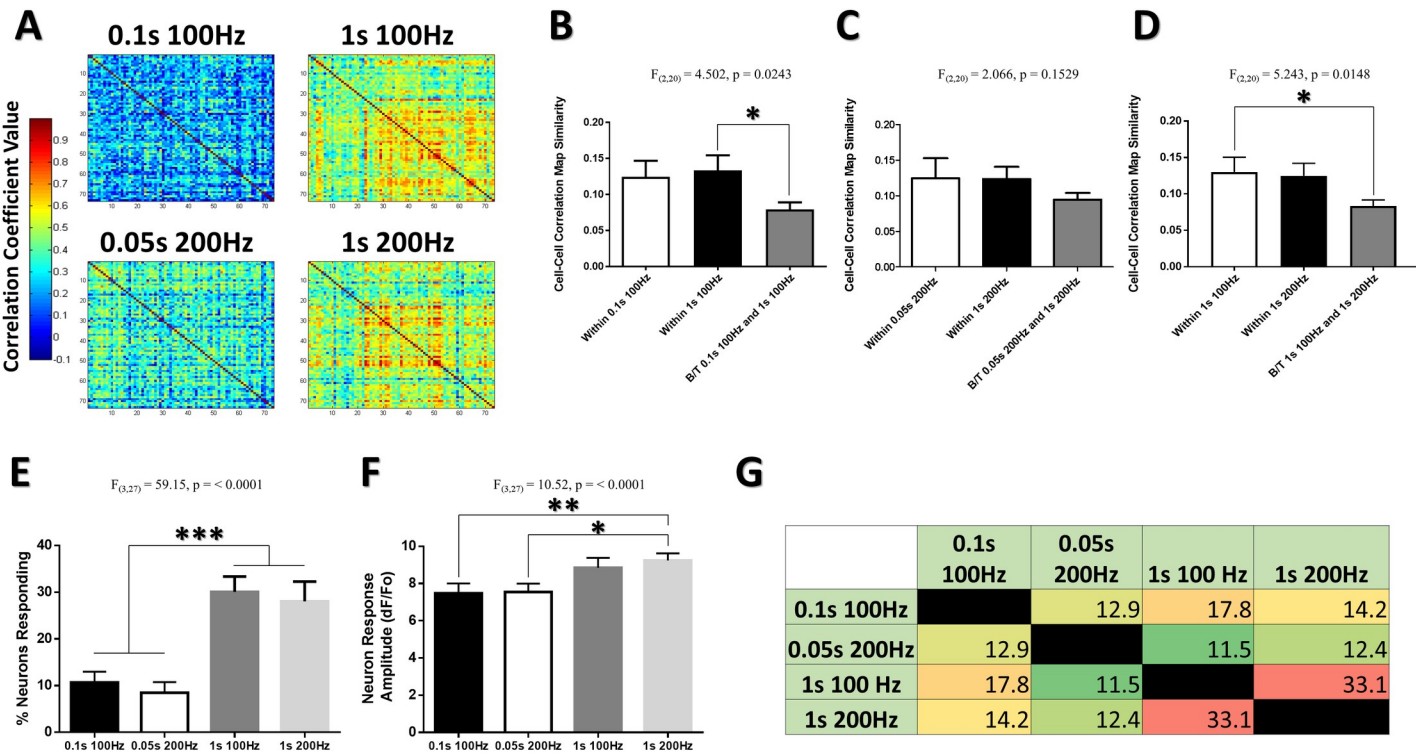

**Fig 6. Longer stimuli evoke more consistent patterns of activity in somatosensory neuronal networks.** (*A*) Example heat map matrices of the pairwise stimulus-evoked correlation coefficient between a population of 74 neurons within the cFL somatosensory cortex of one example animal at 160µm depth for 100 and 200Hz stimuli of different durations. Structural similarity comparison of the maps for all trials depicts greater map similarity for within stimulus conditions than between stimulus for 0.1s 100Hz and 1s 100Hz comparisons (*B*), and a trend for 0.5s 200Hz and 1s 200Hz comparisons (*C*). Significantly less map similarity is seen between 1s 100Hz and 1s 200Hz comparisons than for within stimulus comparisons (*D*). A significant effect of stimulus duration is observed in the percent of neurons responding (*E*) and in the neuron response amplitude (*F*). (*G*) Color coded quantification chart displaying the percentage overlap between neuron populations responsive to each stimulus quantified for all animals (N = 10). *p < 0.05; **p < 0.01; ***p < 0.001.

of our animals studied, the percent of neurons responding, area under the curve, neuron response amplitude, and the Ca$^{2+}$ response decay tau was not statistically different between 0.05s 200Hz and 0.1s 100Hz stimuli (Fig 5E–5H). Thus, the low percent overlap for these 10 oscillation stimuli and the trend towards dissimilarity when comparing their correlation maps (Fig 5A–5D) is a better indicator of differences between the population responses to these 10 oscillation stimuli than the averaged response characteristics of these populations (Fig 5E–5H).

## Longer stimuli evoke more consistent patterns of activity in somatosensory neuronal networks

Within the vibrissal sensory cortex of rodents, the perceived intensity of oscillatory stimuli grows over the course of longer stimulus durations [47]. Fig 5 shows that two short duration stimuli with different frequency, but a fixed absolute number of limb oscillations, activate largely different populations of neurons in the somatosensory cortex. To determine whether the pattern of cortical activity elicited by short duration stimuli would be consistent for longer duration stimuli of the same frequency, short duration 0.1s 100Hz and 0.05s 200Hz stimuli were compared to longer duration 1s 100Hz and 1s 200Hz stimuli, respectively. Fig 6A depicts stimulus-evoked correlation maps from the cFL somatosensory cortex of a representative

animal. The structural similarity index was used to compare stimulus-evoked correlational maps between these stimuli of fixed frequency but different duration across multiple trials within the same animals (see S1 Fig for an example of this). A significant main effect was found for the 100Hz stimulus condition ($F_{(2,20)}$ = 4.502, p = 0.0243), with the "Within 1s 100Hz" condition having significantly greater stimulus-evoked map similarity than the "B/T 0.1s 100Hz and 1s 100Hz" condition in post-hoc comparisons (Fig 6B). This effect was not significant for the 200 Hz stimuli, though a similar pattern is apparent (Fig 6C). Furthermore, a significant main effect is detected when comparing 1s 100Hz to 1s 200Hz stimulus condition ($F_{(2,20)}$ = 5.243, p = 0.0148), with the "Within 1s 100Hz" condition having significantly greater stimulus-evoked map similarity than the "B/T 1s 100Hz and 1s 200Hz" condition in post-hoc comparisons (Fig 6D). Thus, stimulus-evoked correlational map structure was more similar for repeated trials within stimuli than for comparisons between stimuli of different duration. Longer duration stimuli also elicited a greater percentage of neurons responding (Fig 6E), slightly greater average neuron response amplitude (Fig 6F), and more overlap between their responsive cell populations (Fig 6G).

To further examine whether short, high frequency stimuli (0.1s 100Hz and 0.05s 200Hz) lead to different cortical responses than long, low frequency stimuli (1s 3Hz and 10Hz), we compared the stimulus-evoked correlational maps of these stimuli across multiple trials within the same animals (S2A and S2B Fig). A significant main effect was found for the cell-cell correlation map similarity for within stimuli comparisons to between stimuli comparisons ($F_{(1.834,18.34)}$ = 12.53, p = 0.0005), with multiple post-hoc comparisons indicating significantly greater map similarity for within stimulus trials as compared to between stimulus trial comparisons (S2B Fig). These stimuli were not differentiated by the mean neuron response amplitude of their responsive neural populations ($F_{(1.811,16.30)}$ = 1.353, p = 0.2835) (S2C Fig), nor the area under the curve of neural responses ($F_{(1.912,17.21)}$ = 2.372, p = 0.1248) (S2E Fig). A significant main effect was however seen for the percent of neurons responding ($F_{(1.829,16.46)}$ = 5.311, p = 0.0186) and for the neuron decay tau ($F_{(2.517,22.66)}$ = 7.051, p = 0.0025), with 1s 10Hz stimuli showing a higher percent neurons responding than 0.05s 200Hz and greater decay tau than 0.1s 100Hz and 0.05s 200Hz (S2D and S2F Fig). A low percent overlap in the responsive neuronal populations was observed between all stimuli (S2G Fig).

## Discussion

Calcium imaging is a powerful tool to record sensory-evoked response properties of groups of visually identified neurons in the supragranular cortical layers [1,29–32], particularly in rodents. The present study investigated the sensory-evoked $Ca^{2+}$ response in the primary somatosensory cortex of mice to different frequencies and durations of mechanical square-wave limb stimulation using piezoelectric bending actuators. Bending actuators are often used for the study of cortical excitability during disease or after injury due to their simplicity of use and low cost. Despite having no clear naturalistic equivalent, these stimulators likely provide a highly multimodal input to the somatosensory cortex due to their high oscillation amplitude and method of attachment to the limb. We therefore hypothesized that differential activation of diverse populations of peripheral mechanoreceptors for each stimulus frequency would elicit different patterns of activity in somatosensory cortical neuronal networks, and that these distinct patterns would be apparent when analyzing the activity of neurons in these networks. Consistent with previous studies examining the representation of vibro-tactile stimuli in the mouse primary somatosensory cortex [48], our data demonstrates that patterns of suprathreshold (i.e. spiking) activity in the local neuronal network in S1 are modulated by the frequency and duration of our square-wave stimuli. Notably, $Ca^{2+}$ responses to square-wave stimuli do

not scale proportionally to the absolute quantity of limb oscillations within a temporal period. Instead, 3Hz and 10Hz stimuli appear to show similar levels of cortical activation but are distinguished by their dissimilar patterns of activity in the neuronal network. High frequency 100Hz, 200Hz and 300Hz stimuli all elicit greater response strength (number of neurons active, AUC and percent of strongly responsive neurons) than lower frequency 3Hz and 10Hz stimuli. However, the percentage of neurons responding, the response amplitude and AUC of responsive neurons did not scale linearly with stimulus frequency within the high frequency range. Notably, a trend of reduced response strength is even apparent for cortical responses to 300Hz relative to 100 and 200Hz. This non-linear scaling is consistent with findings that neurons responding to high frequency stimuli over 100Hz are not temporally entrained, but are instead tuned to selective features of the high frequency stimulus [23]. A greater average cross-correlation for high frequency stimuli was observed (Fig 4E), potentially due to stronger and more frequent responses within the neural population to these high frequency stimuli. Stimulus-evoked correlation maps were more structurally similar for repeated stimuli of the same frequency as compared to repeated stimuli between different stimulus frequencies, suggesting a degree of representational consistency across stimulus trials within a single imaging session.

The correlation structure for different stimuli are known to be influenced by multiple types of correlation including signal correlation ($r_{signal}$) and spike-count (noise) correlation ($r_{sc}$) [49,50]. Measured $r_{sc}$ has previously been found to be small when neuron responses are weak [49], and the majority of studies measuring $r_{sc}$ within the primate sensory cortex have indicated low $r_{sc}$ in the range of 0.1–0.2 in pairs of nearby neurons with strong $r_{signal}$ [49]. Within the mouse whisker barrel cortex, $r_{sc}$ is highest among neurons with similar orientation tuning and excitability to single-whisker deflections [46]. Recent work examining the representation of vibro-tactile stimuli within the limb associated somatosensory cortex of mice has further supported low $r_{sc}$ measures in pairs of excitatory cells with strong $r_{signal}$, however with higher $r_{sc}$ among pairs of inhibitory interneurons [48]. It has been argued that minimizing correlation within the neural population serves to reduce representational redundancy and improves representational efficiency [51,52]. The stimulus associated cell-cell correlation maps here defined patterns of activity within the network, and demonstrated that these patterns of activity were distinct between stimuli. Although neuronal activity to repeated presentations of the same stimulus is known to vary on a trial by trial basis [48], our data indicate that the structure of the correlation between neuron pairs remains more consistent across trials of the same stimulus than in comparison of trials between different stimuli. However, $Ca^{2+}$ imaging deconvolution methods, that have greater precision when combined with $Ca^{2+}$ indicators of higher temporal fidelity and larger signal-to-noise ratios [53–55], were not used in this study to approximate spike times and compute spike-count correlations for the pairwise comparisons. Furthermore, the sparse and discontinuous nature of the five frequencies of stimulation tested does not give sufficient information to determine precise tuning curves for cells within the network. Stimulus-evoked correlation maps also can not define whether cells displayed correlated $Ca^{2+}$ signals due to sensory-evoked $Ca^{2+}$ transients, or due to neural silence during the stimulus period. Future studies could make use of $Ca^{2+}$ indicators with higher temporal fidelity, such as the GCaMP7F $Ca^{2+}$ indicator [56] or GEVIs [57], to better deconvolve spike timing and compute both signal and spike-count correlations and their contribution to the neuronal representation for each stimulus.

A general feature of hierarchical somatosensory systems holds that activity within neural populations along the afferent pathway from the periphery are organized into systematic body maps according to their somatotopically arranged receptive fields. Within this system, stimulus information is parcellated on a hierarchical network level as relevant features from each previous level are selectively represented within the higher-order representation of the next

level [8]. Although many of the fundamental mechanisms by which afferent signals arise from peripheral mechanoreceptors have been well studied [8,24,58–64], mounting evidence points to a large degree of functional overlap between neurons at multiple levels of the CNS in responding to afferent signals from multiple somatosensory modalities [8,65–69]. Recent research strongly indicates that single cells within S1 cortex and below may already hold high-level representations of activity from multiple mechanosensory modalities, multiple receptive field locations, or even higher order features such as those associated with movement direction, velocity, texture, shape, orientation, compliance, slip, and rolling [8,65,66,69,70]. A proportion of these multimodal neurons have even been found to reside within the spinal cord and brainstem nuclei as the first synaptic targets for afferent fibers emanating from the limbs [24,62,67,70–73]. It has been recently shown in cats that cells of the cuneate nucleus encode high level information related to unique combinations of contact initiation/cessation, slip, and rolling contact [70]. Thus, even proximal neuronal networks in hierarchical somatosensory pathways may represent complex sensory information within their local neuronal population. It is therefore plausible that the limb-associated S1 cortex in rodents already holds representations of complex stimulus features within its single cell and population code similar to higher mammals. The non-linear differences in response strength to our stimuli, and differences in the cortical representation demonstrated by our 10 oscillation stimuli in comparison to each other and to their longer temporal duration equivalents, both support that mouse S1 cortex is not simple entrained by the stimulus frequency but is instead modulated by the frequency and length of the stimulus. Single cells within S1 barrel cortex have been shown to display tuning related to multi-whisker features such as center-surround feature extraction, angular tuning, and multi-whisker correlations [74], thereby lending further weight to the prediction that the limb-associated S1 cortex may also represent convergent multimodal stimulus features within the stimulus representations we have depicted here.

This study did not attempt to directly examine the contribution of specific mechanoreceptor populations in the limbs to the afferent signal generated by our square-wave stimuli and their corresponding cortical responses. While it is not possible to give specific predictions on how strongly each modality was represented within the afferent response and cortical activity, peripheral mechanoreceptors are known to have varied activation thresholds at a range of frequencies [75,76], and the overall pattern of evoked activity would therefore reflect mixed afferent signals generated by the stimuli in this study. Saturation of the calcium response for high frequency limb oscillation could potentially contribute to similarities in the percent of neurons responding (Fig 2E) and the average population response magnitude (Fig 3) to our high frequency stimuli. However, distinct patterns of cellular responses for neurons responsive to each of the high frequency stimuli (Fig 2G), and differences observed in the correlational map structure between different high frequency stimuli (Figs 5 and 6), suggest that the calcium response maintained sufficient fidelity to record distinct representations of stimulus frequency. Future studies could make use of measurement methods with higher temporal fidelity and transgenic animals expressing channelrhodopsins in genetically-identified modality specific afferents to further examine the contribution of particular modalities to the cortical response [23,62]. This study also made use of anesthetized animals and passive stimulation. Anesthesia has previously been shown to reduce the magnitude and spread of cortical activation from simple somatic stimuli [77–79]. Anesthesia also reduces the potential contribution of corollary discharge and reafference due to voluntary action in modulating sensory input and cortical responses [80]. Somatosensory responses are known to be modulated by the relevance of stimuli to behavior and task performance [81–86], and a combination of responses to artificial and simplified stimuli may not fully predict responses of a sensory system to complex naturalistic stimuli [87–90]. Within the rodent barrel cortex, for instance, aperiodic stimulus trains associated with

naturalistic conditions may have sufficient noise to modulate the response magnitude of cortical networks, especially at higher stimulus frequencies [91]. Furthermore, movement-related suppression of self-generated somatosensory activation uses thalamocortical mechanisms in mice to enhance selectivity for touch-related signals [92]. Although the square-wave stimuli in our study lack the aperiodic, broadband, and re-afferent nature of naturalistic stimuli, our results demonstrate that the limb-associated primary somatosensory cortex of mice differentially represents these distinct mechanical limb stimuli at the level of local neuronal networks in the S1 of anesthesized animals and serves to inform future studies about the relative differences in cortical responses expected between the various stimuli observed herein.

## Supporting information

**S1 Fig. Visual example of the structural similarity (SSIM)-based method for trial-by-trial comparisons of correlational map similarity.** In this example depicting the stimulus-evoked correlation maps from an example animal in the cHL at 130um depth, the SSIM value and SSIM comparison image display greater similarity for within trials of 0.1s 100Hz (*A*) and within trials for 1s 100Hz (*B*). Comparisons between trials of stimuli with different frequency or temporal durations demonstrate less structural similarity and a lower SSIM value (*C*). To generate the SSIM charts seen in Figs 5 and 6 and 8 trials of each stimulus were compared for SSIM value within and between stimuli as shown in (*D*) (comparison between 4 trials of each shown here).
(DOCX)

**S2 Fig. Short duration, high frequency stimuli lead to differential cortical activity from long duration, low frequency stimuli.** (*A*) Example heat map matrices of the pairwise stimulus-evoked correlation coefficient between a population of 74 neurons within the cFL somatosensory cortex of one example animal at 160μm depth for 0.1s 100Hz, 0.05s 200Hz, 1s 3Hz, and 1s 10Hz stimuli. Structural similarity comparison of the maps depicts greater map similarity for within stimulus trial conditions than between stimulus trial comparisons (*B*). A significant effect of stimulus is observed in the percent of neurons responding (*D*) and in the neuron decay tau (*F*), but not in the neuron response amplitude (*C*) or the neuron AUC (*E*). (*G*) Color coded quantification chart displaying the percentage overlap between neuron populations responsive to each stimulus quantified for all animals (N = 10). $^*p < 0.05$; $^{**}p < 0.01$; $^{***}p < 0.001$.
(DOCX)

## Author Contributions

**Conceptualization:** Mischa V. Bandet, Bin Dong, Ian R. Winship.

**Data curation:** Ian R. Winship.

**Formal analysis:** Mischa V. Bandet.

**Investigation:** Mischa V. Bandet, Bin Dong.

**Methodology:** Mischa V. Bandet, Bin Dong, Ian R. Winship.

**Project administration:** Bin Dong, Ian R. Winship.

**Resources:** Ian R. Winship.

**Software:** Mischa V. Bandet.

**Supervision:** Ian R. Winship.

Writing – **original draft:** Mischa V. Bandet, Ian R. Winship.

Writing – **review & editing:** Mischa V. Bandet, Ian R. Winship.

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
