## [Decision Letter · Decision Letter 0]

16 Sep 2020

PONE-D-20-21079

Distinct patterns of activity in individual cortical neurons and local networks in primary somatosensory cortex of mice evoked by mechanical limb stimulation

PLOS ONE

Dear Dr. Winship,

Thank you for submitting your manuscript to PLOS ONE. After careful consideration, we feel that it has merit but does not fully meet PLOS ONE’s publication criteria as it currently stands. Therefore, we invite you to submit a revised version of the manuscript that addresses the points raised during the review process.

The manuscript should be seriously revised to provide details and justify some applied methods, address questions regarding statistical analysis, and discuss results in the context of recent publications to address the reviewers' concerns:

These results are not all entirely novel and the authors do not convincingly highlight their importance. The reader is left wondering what exactly this paper tries to convey about cortical coding of somatosensory stimuli.

1. Why do the authors use square wave stimuli instead of pure sinusoids? Sinusoidal stimuli were used in almost all seminal studies on vibrotactile neural coding in the somatosensory periphery and cortex. This therefore makes it problematic to compare the author’s results to those of previous studies. Also, square wave stimuli have harmonics and it is therefore misleading to label a stimulus with a single frequency value (e.g. 300 Hz).

2. It should be discussed how used artificial stimuli are correlated with native ones. For example, it is not clear how physiological are high frequency stimulation at 100, 200, and particularly 300 Hz for the skin of mice? Are rapidly adapted receptors in the skin suitable for discrimination these high frequency stimuli?

3. A recent study by Prsa et al.  (Nature 2019) has performed a similar set of very thorough experiments with regards to the mapping of S1 neurons to forelimb vibrations. While the present manuscript offers data in a slightly different frequency range (3-300 Hz as compared to Prsa et al. ~100Hz – 1kHz) and combines both forelimb and hindlimb stimulation, the authors do little to highlight these differences to emphasize the advance of the present work. Moreover, the present work is done in anesthetized animals under urethane anesthesia whereas Prsa et al. report responses from awake behaving animals; cortical responses are likely heavily impacted by anesthesia, making the interpretability of the stimulus-response properties of the neurons reported here a challenge.

4.     The analysis of single cell tuning to frequency would benefit from being expanded to better support claim 1) in the babstract. The authors declare a cell to be “responding” or “not responding” to a given frequency based on a threshold instead of reporting complete tuning curves to all frequencies. For example, are some neurons tuned to a specific frequency, but have wide tuning curves (i.e., have small but significant responses to frequencies near the preferred frequency) whereas others have narrow tuning bands (i.e,. respond robustly to only one frequency?). Are the neurons that respond to multiple frequencies simply thresholded (i.e., they respond similarly to all stimulations above a given frequency).

a.     The data in Fig 2C are not convincing of an example neuron that responds specifically to a single frequency, and draws into question the robustness of the thresholding approach. Showing tuning curves and parameterizing those tuning curves (preferred frequency and modulation depth) would be preferable.

5.     The correlation matrix analyses are confounded by the fact that the network is overall more active during higher frequency stimulations and thus this analysis is overly complicated for the result that is being shown. The population is more active during high-frequency stimulations, forcing correlations between neurons to be higher during high-frequency stimulations. If some neurons are not recruited by low frequency stimuli, but are for high frequency stimuli, the structure of the correlation map will necessarily change. It is unclear what is gained from the correlation maps over reporting the number/identity of recruited neurons at each frequency, and the latter would be more intuitive. Clustering the correlation matrices and showing that neurons belong to different clusters for different stimulations, or showing that hindlimb/forelimb populations drop out may help bring relevance to the correlation maps.

6. Regarding the stimulus-evoked pair-wise correlation analysis, the authors should carry out a randomization test where they shuffle the stimulus timings and repeat the analysis over a large number of iterations (>1000). The correlation coefficients obtained with the non-shuffled data should then be compared to the confidence intervals of the shuffled data to test for significance. Indeed, the obtained correlation data might have nothing to do with the stimulus but be intrinsic to the neural network or possibly be even noise correlations.

7.     The hindlimb and forelimb populations could be analyzed separately to better characterize each population. For example, perhaps hindlimb neurons are responsive to all frequencies, but forelimb neurons are only responsive to high frequencies. This means that the increase in numbers of neurons active during high frequency stimulation is simply due to the forelimb being sensitive only to high frequencies. Moreover, we might expect the hindlimb neurons to be correlated with one another over the forelimb and vice versa.  

8.     The energy of the stimuli are not properly matched (100Hz stimulations have much more energy than 3Hz stimulations). The authors identify this possible confound that higher frequency simply contain “more stimulation” than lower frequency stimuli. To address this the authors present high frequency, short duration stimuli, but do not compare those to the low frequency, long duration responses. Comparing the responses between short high frequency and long low frequency stimulations would be welcome.

9. The authors claim that all their vibratory stimuli had an amplitude of 1 mm. How exactly was this calibrated? A thorough description of the measurement apparatus and procedure is missing. This is key because it is crucial to demonstrate that the vibration amplitude did not vary across different frequencies. The limb load might not be equally displaced by a 300 Hz vibration than by a 3 Hz vibration. Why was such a high amplitude chosen? Previous studies in rodents used an order of magnitude smaller amplitudes (in the um range) which evoked highly reliable responses and is already well beyond their perceptual thresholds. It therefore seems highly plausible that the 1 mm oscillations evoked highly saturated neural responses, which are definitely not in the natural perceptual range of mice.

10. Very little technical detail is given about the stimulation apparatus and procedure (p.5, lines 112 and 113). How big is the actuator and its end point? How was it driven? At what sampling rate? With what amplifier? Was a sensor used to measure the actual movement that was produced? How was the stimulus integrity assured? Was it consistent across several repetitions?

11. A threshold criterion was used to identify responsive neurons. Why was this preferred to statistical tests? Was this analysis done on the average traces or on every single trial? The authors claim that this procedure was “optimal for separating responsive neurons from noise”, but based on what criteria?

12. Two-photon image processing and determination of responding neurons, Line#165:

Authors indicate that threshold response criteria for Ca2+ fluorescence is 3X increase of standard deviation of fluorescence. However, in Results Line 270 they use another criteria “strongly responsive” neurons in which Ca2+ transient of ΔF/F0 > 10%. The approach for identification of responses should be clarified in Method section.

Major concerns:

- Page 3, lines 62-65: this statement should be nuanced. A limb flexion through a single axis of motion certainly does not recruit all mechanoreceptor types and certainly not equally (it will preferentially recruit proprioceptors).

- Page 3, lines 65-70: it is not clear what the point of this sentence is. What do the authors try to convey here and how does it relate to their experiments?

- The authors use “limb movement” and “limb vibration” interchangeably throughout the manuscript. These are fundamentally different stimuli with each having its own specialized mechanoreceptors (Meissner and Pacinian corpuscles transduce vibrations and proprioceptors transduce limb movement).

- Page 6, line 123: can you give the precise coordinates of the imaging sites and how do these compare to those previously reported for the location of the forelimb and hindlimb S1?

- Page 6, line 132: why is the impedance of the pipette relevant for injections?

- Page 7, line 153: the authors should explain what “photon-shot noise” is.

- Page 8, line 169: a detailed description of the AUC analysis is missing. What exactly is being classified here? What does “half-width” refer to? Where is time-to-peak analyzed in the results?

- Page 8, lines 179-181: was the data z-scored prior to calculating the correlation coefficients?

- Page 9, line 189: a better description of the “seed-based” analysis is needed.

- The stimulus evoked trace in figure 2C does not appear to be an actual calcium transient, but looks more like noise fluctuations. Is this the best available example for a 10 Hz selective neuron? Is this bump that appears in the average trace only due to a single trial or does it reliably appear across multiple trials?

- In general, the authors cannot rule out frequency selective responses because of the limited number of frequencies tested. We do not know what happens beyond 300 Hz.

- Page 12, lines 266-267: could this be due to saturated Ca responses? In general, why is the decay of the calcium transient relevant? What does it reflect exactly?

- Page 14, lines 299-300: or could it be due to the fact that nearby neurons have more similar expression levels?

- Page 14, lines 308-310: how does this compare to analysing the pre- or post- stimulus baseline period and also is it significant when running a randomization test with shuffled data?

- Page 14, lines 312-313: could this simply be due to the fact that higher frequencies have stronger and more frequent responses and thus yield higher correlation coefficients? Was Z scoring performed?

- Page 15, lines 340-342: It is not clear what this means or implies.

- Page 16, lines 351-352: the authors compare 0.1 s and 0.05 s stimulus durations with 1 s durations (a 10 to 20 fold duration difference). Why such big differences and such short stimuli? A 50 ms vibration will hardly evoke any spikes.

- Page 16, lines 359-360: how do the authors explain this?

- In 2BC, labels for the frequency of stimulation would be helpful.

-     The analysis of the population dF/F (Fig 3A) does not provide much insight. While not incorrect, there is little gain in understanding of encoding properties by showing the population as a whole is more active when the authors have shown in the previous figure that the number of neurons recruited is increased for higher frequency stimuli.  The amplitude, decay time and AUC are all strongly related to each other, as calcium signals are subject to decay. It is unclear what the additional functional significance is of the AUC or decay time being different across stimulation frequencies.

-     The choice of a square wave pulse for the stimulation is a minor caveat, as the abrupt changes in limb direction at the onset and offset of the square wave will cause high frequency ringing/vibration of the limb that will depend on the intrinsic characteristics of the limb.

We look forward to receiving your revised manuscript.

Kind regards,

Gennady Cymbalyuk, Ph.D.

Academic Editor

PLOS ONE

Journal Requirements:

2. To comply with PLOS ONE submissions requirements, please provide methods of sacrifice in the Methods section of your manuscript.

Reviewers' comments:

Reviewer's Responses to Questions

**Comments to the Author**

1. Is the manuscript technically sound, and do the data support the conclusions?

Reviewer #1: Partly

Reviewer #2: Yes

Reviewer #3: Partly

2. Has the statistical analysis been performed appropriately and rigorously? 

Reviewer #1: No

Reviewer #2: Yes

Reviewer #3: Yes

3. Have the authors made all data underlying the findings in their manuscript fully available?

Reviewer #1: Yes

Reviewer #2: Yes

Reviewer #3: Yes

4. Is the manuscript presented in an intelligible fashion and written in standard English?

Reviewer #1: Yes

Reviewer #2: Yes

Reviewer #3: Yes

5. Review Comments to the Author

Reviewer #1: In their study, Bandet et al. use calcium imaging of neuronal populations in the mouse somatosensory cortex to show that distinct vibrotactile stimuli are represented with distinct patterns of neuronal activation. They report that somatosensory cortex neurons distinguish between different frequencies of forelimb and hindlimb stimulation in the following manner:

- The number of neurons responding to high frequency (100 Hz, 200 Hz and 300 Hz) stimuli is greater than the number of neurons responding to low frequency (3 Hz and 10 Hz) stimuli (Fig. 1E).

- The response amplitude of neurons (i.e. size of calcium transients) is independent of stimulus frequency.

- High frequency stimuli evoke longer decays of calcium transients and AUC values, but these two measures do not scale linearly with frequency.

- Pair wise stimulus evoked correlations between neurons varied between different stimulus frequencies. Higher frequency stimuli had higher correlations.

- Longer duration stimuli evoke more consistent activity patterns than shorter stimuli.

These results are not all entirely novel and the authors do not convincingly highlight their importance. The reader is left wondering what exactly this paper tries to convey about cortical coding of somatosensory stimuli.

There are several important issues and questions that the authors should address. These are listed here below.

1. Why do the authors use square wave stimuli instead of pure sinusoids? Sinusoidal stimuli were used in almost all seminal studies on vibrotactile neural coding in the somatosensory periphery and cortex. This therefore makes it problematic to compare the author’s results to those of previous studies. Also, square wave stimuli have harmonics and it is therefore misleading to label a stimulus with a single frequency value (e.g. 300 Hz).

2. The authors claim that all their vibratory stimuli had an amplitude of 1 mm. How exactly was this calibrated? A thorough description of the measurement apparatus and procedure is missing. This is key because it is crucial to demonstrate that the vibration amplitude did not vary across different frequencies. The limb load might not be equally displaced by a 300 Hz vibration than by a 3 Hz vibration. Why was such a high amplitude chosen? Previous studies in rodents used an order of magnitude smaller amplitudes (in the um range) which evoked highly reliable responses and is already well beyond their perceptual thresholds. It therefore seems highly plausible that the 1 mm oscillations evoked highly saturated neural responses, which are definitely not in the natural perceptual range of mice.

3. Very little technical detail is given about the stimulation apparatus and procedure (p.5, lines 112 and 113). How big is the actuator and its end point? How was it driven? At what sampling rate? With what amplifier? Was a sensor used to measure the actual movement that was produced? How was the stimulus integrity assured? Was it consistent across several repetitions? Etc…

4. A threshold criterion was used to identify responsive neurons. Why was this preferred to statistical tests? Was this analysis done on the average traces or on every single trial? The authors claim that this procedure was “optimal for separating responsive neurons from noise”, but based on what criteria?

5. Regarding the stimulus-evoked pair-wise correlation analysis, the authors should carry out a randomization test where they shuffle the stimulus timings and repeat the analysis over a large number of iterations (>1000). The correlation coefficients obtained with the non-shuffled data should then be compared to the confidence intervals of the shuffled data to test for significance. Indeed, the obtained correlation data might have nothing to do with the stimulus but be intrinsic to the neural network or possibly be even noise correlations.

Additional comments/questions:

- Page 3, lines 62-65: this statement should be nuanced. A limb flexion through a single axis of motion certainly does not recruit all mechanoreceptor types and certainly not equally (it will preferentially recruit proprioceptors).

- Page 3, lines 65-70: it is not clear what the point of this sentence is. What do the authors try to convey here and how does it relate to their experiments?

- The authors use “limb movement” and “limb vibration” interchangeably throughout the manuscript. These are fundamentally different stimuli with each having its own specialized mechanoreceptors (Meissner and Pacinian corpuscles transduce vibrations and proprioceptors transduce limb movement).

- Page 6, line 123: can you give the precise coordinates of the imaging sites and how do these compare to those previously reported for the location of the forelimb and hindlimb S1?

- Page 6, line 132: why is the impedance of the pipette relevant for injections?

- Page 7, line 153: the authors should explain what “photon-shot noise” is.

- Page 8, line 169: a detailed description of the AUC analysis is missing. What exactly is being classified here? What does “half-width” refer to? Where is time-to-peak analyzed in the results?

- Page 8, lines 179-181: was the data z-scored prior to calculating the correlation coefficients?

- Page 9, line 189: a better description of the “seed-based” analysis is needed.

- The stimulus evoked trace in figure 2C does not appear to be an actual calcium transient, but looks more like noise fluctuations. Is this the best available example for a 10 Hz selective neuron? Is this bump that appears in the average trace only due to a single trial or does it reliably appear across multiple trials?

- In general, the authors cannot rule out frequency selective responses because of the limited number of frequencies tested. We do not know what happens beyond 300 Hz.

- Page 12, lines 266-267: could this be due to saturated Ca responses? In general, why is the decay of the calcium transient relevant? What does it reflect exactly?

- Page 14, lines 299-300: or could it be due to the fact that nearby neurons have more similar expression levels?

- Page 14, lines 308-310: how does this compare to analysing the pre- or post- stimulus baseline period and also is it significant when running a randomization test with shuffled data?

- Page 14, lines 312-313: could this simply be due to the fact that higher frequencies have stronger and more frequent responses and thus yield higher correlation coefficients? Was Z scoring performed?

- Page 15, lines 340-342: It is not clear what this means or implies.

- Page 16, lines 351-352: the authors compare 0.1 s and 0.05 s stimulus durations with 1 s durations (a 10 to 20 fold duration difference). Why such big differences and such short stimuli? A 50 ms vibration will hardly evoke any spikes.

- Page 16, lines 359-360: how do the authors explain this?

Reviewer #2: The paper entitled: “Distinct patterns of activity in individual cortical neurons and local networks in primary somatosensory cortex of mice evoked by mechanical limb stimulation” is dedicated to the very important problem of encoding at the level of the somatosensory cortex. This work is important because for now mechanisms and principles of specificity and qualitative separation of sensory signals are not elucidated enough. Authors used very sensitive method of two-photon microscopy for visualization of neuronal activation in vivo for study of cortical networks involved in encoding of mechanical stimuli. Approach is appropriate, well developed data collection and statistical analyses makes the results rigorous and argumentative.

The manuscript could be accepted for publication.

I just would like to add few minor comments, which, in my opinion, could improve it.

1). Two-photon image processing and determination of responding neurons, Line#165:

Authors indicate that threshold response criteria for Ca2+ fluorescence is 3X increase of standard deviation of fluorescence. However, in Results Line 270 they use another criteria “strongly responsive” neurons in which Ca2+ transient of ΔF/F0 > 10%. The approach for identification of responses should be clarified in Method section.

2). It should be discussed how used artificial stimuli are correlated with native ones. For example, it is not clear how physiological are high frequency stimulation at 100, 200, and particularly 300 Hz for the skin of mice? Are rapidly adapted receptors in the skin suitable for discrimination these high frequency stimuli?

Reviewer #3: In this manuscript, Bandet et al. characterize the response properties of S1 cortical neurons to mechanical vibration of the forelimb and hindlimb using Ca2+ imaging. Unfortunately, a recent study by Prsa et al. (Nature 2019) has performed a similar set of very thorough experiments with regards to the mapping of S1 neurons to forelimb vibrations. While the present manuscript offers data in a slightly different frequency range (3-300 Hz as compared to Prsa et al. ~100Hz – 1kHz) and combines both forelimb and hindlimb stimulation, the authors do little to highlight these differences to emphasize the advance of the present work. Moreover, the present work is done in anesthetized animals under urethane anesthesia whereas Prsa et al. report responses from awake behaving animals; cortical responses are likely heavily impacted by anesthesia, making the interpretability of the stimulus-response properties of the neurons reported here a challenge. That said, the data collected are of high quality and technically sound, and the manuscript could be improved with some revisions to the analyses listed below.

The abstract states the main results to be: 1) S1 neurons fall into 2 categories: frequency specific responses, or responsive to multiple frequencies 2) High frequency stimulation recruits more of the population, both in number of cells recruited and the magnitude of each cell’s activity. 3) Taken as a population, the pattern of activities are unique for individual stimuli.

Major concerns:

1) The analysis of single cell tuning to frequency would benefit from being expanded to better support claim 1) in the babstract. The authors declare a cell to be “responding” or “not responding” to a given frequency based on a threshold instead of reporting complete tuning curves to all frequencies. For example, are some neurons tuned to a specific frequency, but have wide tuning curves (i.e., have small but significant responses to frequencies near the preferred frequency) whereas others have narrow tuning bands (i.e,. respond robustly to only one frequency?). Are the neurons that respond to multiple frequencies simply thresholded (i.e., they respond similarly to all stimulations above a given frequency).

a. The data in Fig 2C are not convincing of an example neuron that responds specifically to a single frequency, and draws into question the robustness of the thresholding approach. Showing tuning curves and parameterizing those tuning curves (preferred frequency and modulation depth) would be preferable.

2) The correlation matrix analyses are confounded by the fact that the network is overall more active during higher frequency stimulations and thus this analysis is overly complicated for the result that is being shown. The population is more active during high-frequency stimulations, forcing correlations between neurons to be higher during high-frequency stimulations. If some neurons are not recruited by low frequency stimuli, but are for high frequency stimuli, the structure of the correlation map will necessarily change. It is unclear what is gained from the correlation maps over reporting the number/identity of recruited neurons at each frequency, and the latter would be more intuitive. Clustering the correlation matrices and showing that neurons belong to different clusters for different stimulations, or showing that hindlimb/forelimb populations drop out may help bring relevance to the correlation maps.

3) The hindlimb and forelimb populations could be analyzed separately to better characterize each population. For example, perhaps hindlimb neurons are responsive to all frequencies, but forelimb neurons are only responsive to high frequencies. This means that the increase in numbers of neurons active during high frequency stimulation is simply due to the forelimb being sensitive only to high frequencies. Moreover, we might expect the hindlimb neurons to be correlated with one another over the forelimb and vice versa.

4) The energy of the stimuli are not properly matched (100Hz stimulations have much more energy than 3Hz stimulations). The authors identify this possible confound that higher frequency simply contain “more stimulation” than lower frequency stimuli. To address this the authors present high frequency, short duration stimuli, but do not compare those to the low frequency, long duration responses. Comparing the responses between short high frequency and long low frequency stimulations would be welcome.

Minor concerns:

1) In 2BC, labels for the frequency of stimulation would be helpful.

2) The analysis of the population dF/F (Fig 3A) does not provide much insight. While not incorrect, there is little gain in understanding of encoding properties by showing the population as a whole is more active when the authors have shown in the previous figure that the number of neurons recruited is increased for higher frequency stimuli. The amplitude, decay time and AUC are all strongly related to each other, as calcium signals are subject to decay. It is unclear what the additional functional significance is of the AUC or decay time being different across stimulation frequencies.

3) The choice of a square wave pulse for the stimulation is a minor caveat, as the abrupt changes in limb direction at the onset and offset of the square wave will cause high frequency ringing/vibration of the limb that will depend on the intrinsic characteristics of the limb.

6. PLOS authors have the option to publish the peer review history of their article (what does this mean?). If published, this will include your full peer review and any attached files.

Reviewer #1: No

Reviewer #2: No

Reviewer #3: No

---

## [Author Response · Author response to Decision Letter 0]

8 Dec 2020

We thank the reviewers for their helpful comments of our manuscript. Below, we address comments from all reviewers.

Comment: These results are not all entirely novel and the authors do not convincingly highlight their importance. The reader is left wondering what exactly this paper tries to convey about cortical coding of somatosensory stimuli.

Response: This study does not aim to define cortical coding of any particular somatosensory modality or specific aspects of individual stimuli. Instead, our data demonstrate that complex sensory information resulting from the mixed activation of overlapping somatosensory modalities during an artificial square-wave mechanical stimulus can still be differentially represented within the somatosensory cortex of mice. While the square wave mechanical stimulus is not a naturalistic stimulus, different frequencies of stimulation contain overlapping information. Moreover, characterization of the distinct patterns of neural activity elicited by such stimuli is of importance as these forms of square-wave stimulation are used in multiple fields outside of the study of sensory coding, such as in the observation of cortical recovery from damage or disease. We have amended the abstract as follows to highlight this:

Artificial forms of mechanical limb stimulation are used within multiple fields of study to determine the level of cortical excitability and to map the trajectory of neuronal recovery from cortical damage or disease. Square-wave mechanical or electrical stimuli are often used in these studies despite having no clear naturalistic equivalent. To distinguish between somatic stimuli, the primary somatosensory cortex should process dissimilar stimuli with distinct patterns of activity, but must also be able to represent distinct but overlapping stimuli with unique patterns of activity. Here, flavoprotein autofluorescence imaging was used to map the somatosensory cortex of anaesthetized C57BL/6 mice, and in vivo two-photon Ca2+ imaging was used to define patterns of neuronal activation during mechanical square-wave stimulation of the contralateral forelimb or hindlimb at various frequencies (3, 10, 100, 200, and 300 Hz). The data revealed that neurons within the limb associated somatosensory cortex responding to various frequencies of square-wave stimuli exhibit stimulus-specific patterns of activity. Subsets of neurons were found to have sensory-evoked activity that is either primarily responsive to single stimulus frequencies or broadly responsive to multiple frequencies of limb stimulation. High frequency stimuli were shown to elicit more population activity, with a greater percentage of the population responding and greater percentage of cells with high amplitude responses. Stimulus-evoked cell-cell correlations within these neuronal networks varied as a function of frequency of stimulation, such that each stimulus elicited a distinct pattern that was more consistent across multiple trials of the same stimulus compared to trials at different frequencies of stimulation. The variation in cortical response to different square-wave stimuli can thus be represented by the population pattern of supra-threshold Ca2+ transients, the magnitude and temporal properties of the evoked activity, and the structure of the stimulus-evoked correlation between neurons.

Comment: 1. Why do the authors use square wave stimuli instead of pure sinusoids? Sinusoidal stimuli were used in almost all seminal studies on vibrotactile neural coding in the somatosensory periphery and cortex. This therefore makes it problematic to compare the author’s results to those of previous studies. Also, square wave stimuli have harmonics and it is therefore misleading to label a stimulus with a single frequency value (e.g. 300 Hz).

Response: The square-wave stimuli using piezoelectric bending actuators under study have been used in multiple fields outside of the study of sensory coding. As the cortical response to square-wave stimuli used within these previous studies has not been characterized, questions remain on the magnitude of cortical responses expected from mechanical stimuli of this type, whether the somatosensory cortex can differentially respond to these stimuli, and if cortical responses to these stimuli is altered with cortical damage or disease. We agree with the reviewer that the results herein are not directly comparable to those of previous studies using sinusoidal stimuli nor to experimental studies utilizing awake behaving animals. It is however of importance to determine whether the square-wave stimuli used herein are a form of stimulation that the somatosensory cortex can differentially encode despite having no naturalistic equivalents. We have amended text in the introduction as follows:

A better understanding of how the somatosensory system responds to artificial forms of mechanical stimuli, and can distinguish between distinct forms of these stimuli, is important in interpreting studies that use these stimuli as measures of cortical excitability and plastic changes during disease or after injury. Whereas considerable literature exists for somatosensation within the healthy brain in non-human primates (for review, see [1,2]), literature on the somatosensory system of rodents is more limited. Given that rodents are the most prominent animal model used to study reaching and upper limb use and disability[3], it is important to understand how different forms of somatic and movement related stimuli used in rodent research are processed in the rodent somatosensory cortex. Most current research on the limb-associated somatosensory cortex in rodents utilize artificial forms of mechanical[4–7] or electrical stimuli[8,9] to assess sensory-evoked responses using electrophysiology of individual cells or aggregate responses from large cortical regions. Unfortunately, stimulus parameters used in these studies often vary from one study to the next, and few studies offer well defined rationale for their choice of stimuli, thereby making comparisons and interpretation across studies more challenging. Notably, artificial stimuli used for rodent research that may be thought of as relatively simple in comparison to naturalistic stimuli or movements, such as limb oscillation through a single axis of motion, still likely result in the generation of complex multimodal sensory information in the periphery. These multimodal signals may arise from activation of mechanoreceptors at the stimulator attachment site, the propagation of vibration at the fundamental and harmonic frequencies to distal Pacinian corpuscles closely associated with joints and bones[10], the activation of proprioceptive receptors within the muscles and joints, and to a large potential array of signals arising due to deflections of hair follicle associated mechanoreceptors (for review, see [11]). Although particular subsets of cutaneous and proprioceptive mechanoreceptors have long been known in non-human primates to be tuned to characteristics such as stimulus intensity, frequency and receptive field location[1,2,12–15], the harmonics resulting from square-wave oscillation of the entire forelimb or hindlimb of mice would likely elicit highly mixed activity in mechanoreceptor populations, with weighted preference to mechanoreceptor populations most sensitive to the particular frequencies and biomechanics of the stimulus.

Comment: 2. It should be discussed how used artificial stimuli are correlated with native ones. For example, it is not clear how physiological are high frequency stimulation at 100, 200, and particularly 300 Hz for the skin of mice? Are rapidly adapted receptors in the skin suitable for discrimination these high frequency stimuli?

Response: The square-wave stimuli used in this study do not have a closely related equivalent in nature, however they are highly multimodal due to their large magnitude and method of attachment to the limb. The mechanical bending actuators used in this study shakes the entire limb of the mouse at a supraphysical level (>170um oscillation), and would therefore activate not only the exquisitely sensitive Pacinian corpuscles closely associated with the bones and sparsely in the phalanges in mice, but also the slowly adapting and rapidly adapting receptors found at the stimulator attachment site, proprioceptive receptors associated with the muscles and joints, as well as slowly and rapidly adapting receptors associated with hair follicles of the upper arm and shoulder. We can not define the contribution of each of these receptor types within this study, as indicated within the discussion. The premise of the paper is that the somatosensory cortex is able to differentially represent the stimuli despite the supraphysical stimulus level and mixed receptor activation, and the importance in the relative level of cortical activity elicited by each of the stimuli in informing future research that makes use of similar forms of mechanical stimulation under similar experimental conditions. This is now more clearly indicated within the introduction:

A better understanding of how the somatosensory system responds to artificial forms of mechanical stimuli, and can distinguish between distinct forms of these stimuli, is important in interpreting studies that use these stimuli as measures of cortical excitability and plastic changes during disease or after injury. Whereas considerable literature exists for somatosensation within the healthy brain in non-human primates (for review, see [1,2]), literature on the somatosensory system of rodents is more limited. Given that rodents are the most prominent animal model used to study reaching and upper limb use and disability[3], it is important to understand how different forms of somatic and movement related stimuli used in rodent research are processed in the rodent somatosensory cortex. Most current research on the limb-associated somatosensory cortex in rodents utilize artificial forms of mechanical[4–7] or electrical stimuli[8,9] to assess sensory-evoked responses using electrophysiology of individual cells or aggregate responses from large cortical regions. Unfortunately, stimulus parameters used in these studies often vary from one study to the next, and few studies offer well defined rationale for their choice of stimuli, thereby making comparisons and interpretation across studies more challenging. Notably, artificial stimuli used for rodent research that may be thought of as relatively simple in comparison to naturalistic stimuli or movements, such as limb oscillation through a single axis of motion, still likely result in the generation of complex multimodal sensory information in the periphery. These multimodal signals may arise from activation of mechanoreceptors at the stimulator attachment site, the propagation of vibration at the fundamental and harmonic frequencies to distal Pacinian corpuscles closely associated with joints and bones[10], the activation of proprioceptive receptors within the muscles and joints, and to a large potential array of signals arising due to deflections of hair follicle associated mechanoreceptors (for review, see [11]). Although particular subsets of cutaneous and proprioceptive mechanoreceptors have long been known in non-human primates to be tuned to characteristics such as stimulus intensity, frequency and receptive field location[1,2,12–15], the harmonics resulting from square-wave oscillation of the entire forelimb or hindlimb of mice would likely elicit highly mixed activity in mechanoreceptor populations, with weighted preference to mechanoreceptor populations most sensitive to the particular frequencies and biomechanics of the stimulus.

Comment: 3. A recent study by Prsa et al. (Nature 2019) has performed a similar set of very thorough experiments with regards to the mapping of S1 neurons to forelimb vibrations. While the present manuscript offers data in a slightly different frequency range (3-300 Hz as compared to Prsa et al. ~100Hz – 1kHz) and combines both forelimb and hindlimb stimulation, the authors do little to highlight these differences to emphasize the advance of the present work. Moreover, the present work is done in anesthetized animals under urethane anesthesia whereas Prsa et al. report responses from awake behaving animals; cortical responses are likely heavily impacted by anesthesia, making the interpretability of the stimulus-response properties of the neurons reported here a challenge.

Response: The recent study by Prsa et al., 2019 is a true tour-de-force in disseminating cortical responses to high frequency stimuli. Our study does not seek to directly expand on their work, but instead offers a characterization of how S1 neurons respond to artificial square-wave oscillations of the entire forelimb or hindlimb. We agree with the reviewer that the use of anesthetized animals is a limitation that limits the interpretation of the results primarily to other studies using a similar set of experimental conditions. Note that this study does not seek to define somatosensory encoding within naturalistic environments, but is instead a characterization of cortical responses to a range of square-wave stimuli that may be used in other studies examining somatosensory recovery following CNS perturbation and/or injury. It is therefore not fundamental insights into the nature of sensory encoding of tactile stimuli that is characterized within this manuscript, but rather the relative differences in the cortical activation to these stimuli and the ability of the anesthetized mouse somatosensory cortex to differentially respond to these square-wave stimuli that is of importance.

Comments: 4. The analysis of single cell tuning to frequency would benefit from being expanded to better support claim 1) in the abstract. The authors declare a cell to be “responding” or “not responding” to a given frequency based on a threshold instead of reporting complete tuning curves to all frequencies. For example, are some neurons tuned to a specific frequency, but have wide tuning curves (i.e., have small but significant responses to frequencies near the preferred frequency) whereas others have narrow tuning bands (i.e,. respond robustly to only one frequency?). Are the neurons that respond to multiple frequencies simply thresholded (i.e., they respond similarly to all stimulations above a given frequency).

AND

a. The data in Fig 2C are not convincing of an example neuron that responds specifically to a single frequency, and draws into question the robustness of the thresholding approach. Showing tuning curves and parameterizing those tuning curves (preferred frequency and modulation depth) would be preferable.

Response: We have amended Fig 2B,C with two example neurons that demonstrated above threshold responses to all stimuli with preferential activation by high frequency stimuli in B, and strongest activation by the 10Hz stimulus in C. The threshold criterion for a responding or non-responding neuron is necessary with OGB-1 calcium indicator to distinguish calcium transients from noise. Using the threshold criterion, approximately 33% of neurons respond only to a single stimulus (Fig 2F), with the majority of those belonging to the population of neurons responding to 100, 200, or 300Hz stimuli.

Comment: 5. The correlation matrix analyses are confounded by the fact that the network is overall more active during higher frequency stimulations and thus this analysis is overly complicated for the result that is being shown. The population is more active during high-frequency stimulations, forcing correlations between neurons to be higher during high-frequency stimulations. If some neurons are not recruited by low frequency stimuli, but are for high frequency stimuli, the structure of the correlation map will necessarily change. It is unclear what is gained from the correlation maps over reporting the number/identity of recruited neurons at each frequency, and the latter would be more intuitive. Clustering the correlation matrices and showing that neurons belong to different clusters for different stimulations, or showing that hindlimb/forelimb populations drop out may help bring relevance to the correlation maps.

Response: It is expected that higher frequency stimuli will lead to more cortical activity and therefore greater average correlation among the deltaF/Fo neurons within that population as a whole, as was observed in this study. We have added additional Fig 4c,d utilizing a stationary bootstrapping procedure to further illustrate that the significant pairwise correlations within a population varies as a function of the stimulus, a result that is further expanded on in Fig 5 & 6.

Comment: 6. Regarding the stimulus-evoked pair-wise correlation analysis, the authors should carry out a randomization test where they shuffle the stimulus timings and repeat the analysis over a large number of iterations (>1000). The correlation coefficients obtained with the non-shuffled data should then be compared to the confidence intervals of the shuffled data to test for significance. Indeed, the obtained correlation data might have nothing to do with the stimulus but be intrinsic to the neural network or possibly be even noise correlations.

Response: We would like to thank the reviewer for this suggestion. We have used a stationary bootstrap to create figure 4c,d that displays correlation values that were greater than the 97.5th percentile of the bootstrap. A 5 frame mean block length for the stationary bootstrap was used as this corresponds to a block length of 200ms for our 25Hz sampling rate, the minimum period of time between action potentials necessary to detect a calcium transient using OGB-1 (Tada et al., 2014). We have amended the following text within the manuscript:

Methods

A stationary bootstrapping procedure was further used as a test of the significance of the Pearson’s r calculated between the ΔF/Fo of each neuron pair. Within the stationary bootstrapping procedure, 10000 iterations were run and a mean block length of 5 frames (200ms) was used based on previous studies indicating the minimum period of time between action potentials necessary to detect a calcium transient using OGB-1[30]. Pairwise r values greater than the 97.5th percentile of the stationary bootstrap are indicated by light blue pixels (Fig 4c) and the correlation map replotted to show only those correlations meeting the bootstrap criteria (Fig 4D).

Results

We further confirmed the significance of these pairwise cross correlations by running a stationary bootstrapping procedure (see methods). Bootstrapping further supported that the pairwise comparisons meeting the bootstrapping criteria varied as a function of the stimulus (Fig 4c,d). 

Comment: 7. The hindlimb and forelimb populations could be analyzed separately to better characterize each population. For example, perhaps hindlimb neurons are responsive to all frequencies, but forelimb neurons are only responsive to high frequencies. This means that the increase in numbers of neurons active during high frequency stimulation is simply due to the forelimb being sensitive only to high frequencies. Moreover, we might expect the hindlimb neurons to be correlated with one another over the forelimb and vice versa. 

Response: Note that when hindlimb stimulation was applied we recorded only from the hindlimb somatosensory region. When the forelimb was stimulated, we recorded within the forelimb somatosensory region. Hindlimb and forelimb regions display the same patterns characterized within this manuscript, and were lumped together in analysis. Text within the methods has been added as follows:

While imaging the cHL somatosensory region mapped via FA imaging, only the cHL was stimulated. Likewise, only the cFL was stimulated during imaging of the cFL somatosensory region.

Comment: 8. The energy of the stimuli are not properly matched (100Hz stimulations have much more energy than 3Hz stimulations). The authors identify this possible confound that higher frequency simply contain “more stimulation” than lower frequency stimuli. To address this the authors present high frequency, short duration stimuli, but do not compare those to the low frequency, long duration responses. Comparing the responses between short high frequency and long low frequency stimulations would be welcome.

Response: We have added additional figure S2 and added additional results within the manuscript:

To further examine whether short, high frequency stimuli (0.1s 100Hz and 0.05s 200Hz) lead to different cortical responses than long, low frequency stimuli (1s 3Hz and 10Hz), we compared the stimulus-evoked correlational maps of these stimuli across multiple trials within the same animals (Fig S2a,b). A significant main effect was found for the cell-cell correlation map similarity for within stimuli comparisons to between stimuli comparisons (F(1.834,18.34) = 12.53, p = 0.0005), with multiple post-hoc comparisons indicating significantly greater map similarity for within stimulus trials as compared to between stimulus trial comparisons (Fig S2b). These stimuli were not differentiated by the mean neuron response amplitude of their responsive neural populations (F(1.811,16.30) = 1.353, p = 0.2835) (Fig S2c), nor the area under the curve of neural responses (F(1.912,17.21) = 2.372, p = 0.1248) (Fig S2e). A significant main effect was however seen for the percent of neurons responding (F(1.829,16.46) = 5.311, p = 0.0186) and for the neuron decay tau (F(2.517,22.66) = 7.051, p = 0.0025), with 1s 10Hz stimuli showing a higher percent neurons responding than 0.05s 200Hz and greater decay tau than 0.1s 100Hz and 0.05s 200Hz (Fig S2d,f). A low percent overlap in the responsive neuronal populations was observed between all stimuli (Fig S2g).

Comment: 9. The authors claim that all their vibratory stimuli had an amplitude of 1 mm. How exactly was this calibrated? A thorough description of the measurement apparatus and procedure is missing. This is key because it is crucial to demonstrate that the vibration amplitude did not vary across different frequencies. The limb load might not be equally displaced by a 300 Hz vibration than by a 3 Hz vibration. Why was such a high amplitude chosen? Previous studies in rodents used an order of magnitude smaller amplitudes (in the um range) which evoked highly reliable responses and is already well beyond their perceptual thresholds. It therefore seems highly plausible that the 1 mm oscillations evoked highly saturated neural responses, which are definitely not in the natural perceptual range of mice.

and

10. Very little technical detail is given about the stimulation apparatus and procedure (p.5, lines 112 and 113). How big is the actuator and its end point? How was it driven? At what sampling rate? With what amplifier? Was a sensor used to measure the actual movement that was produced? How was the stimulus integrity assured? Was it consistent across several repetitions?

Response: There was no calibration of the stimulator amplitude, and it therefore varied by frequency. A sensor was not used to measure the movement that was produced during these experiments, however we have subsequently imaged the limb oscillation and amended the text with further information on the stimulator used and the stimulus amplitude by frequency. The stimulus amplitude remained consistent across several repetitions of trials within each frequency, but differed between frequencies. We have amended the methods with the following:

Custom-made piezoceramic mechanical bending actuators were used to elicit vibrotactile limb stimulation during FA and Ca2+ imaging. The piezo bending actuators comprised a piezo element (Piezo Systems # Q220-A4-203YB) attached to an electrically insulated metal shaft holding a metal U-bend at its end. The palm of the mouse paw was placed within the U-bend, and the U-bend bent to shape to lightly secure the palm. The metal U-bend made contact across an area of approximately 3x1mm on the palmar and dorsal surface of the hand. Stimulators were driven with square-wave signals from an A-M Systems Model 2100 Isolated Pulse Stimulator. During stimulation, the entire limb underwent an oscillation with the following peak-peak amplitudes by frequency: 280um (3 & 10Hz), 335um (100Hz), 220um (200Hz), 170um (300Hz). 

Comment: 11. A threshold criterion was used to identify responsive neurons. Why was this preferred to statistical tests? Was this analysis done on the average traces or on every single trial? The authors claim that this procedure was “optimal for separating responsive neurons from noise”, but based on what criteria?

Response: The threshold was set based on the signal to noise ratio of the recording. A threshold of three times the standard deviation of the baseline deltaF/Fo excludes noise from being identified as significant responses to the stimuli. This acts as a statistical and repeatable means to exclude fluorescent fluctuations that did not surpass the threshold signal-to-noise ratio for a period of time relevant to the imaging speed and OGB-1 indicator dynamics.

Comment: 12. Two-photon image processing and determination of responding neurons, Line#165: Authors indicate that threshold response criteria for Ca2+ fluorescence is 3X increase of standard deviation of fluorescence. However, in Results Line 270 they use another criteria “strongly responsive” neurons in which Ca2+ transient of ΔF/F0 > 10%. The approach for identification of responses should be clarified in Method section.

Response: We have amended the methods section to further clarify as follows:

A threshold criteria requiring the neuronal Ca2+ fluorescence to increase by 3X the standard deviation of the baseline period ΔF/Fo (baseline defined as 1s before stimulus onset), and remain above this criteria for 160ms (4 successive frames), was found to be optimal for separating responsive neurons from random noise fluctuations. The percentage of neurons that met an additional criteria of peak ΔF/ Fo greater than 10% were deemed “strongly responsive” neurons (Fig 3e) based on previous studies associating >10% increases in OGB-1 ΔF/ Fo with firing of multiple action potentials underlying the Ca2+ response[22].

Comment: - Page 3, lines 62-65: this statement should be nuanced. A limb flexion through a single axis of motion certainly does not recruit all mechanoreceptor types and certainly not equally (it will preferentially recruit proprioceptors).

and

- Page 3, lines 65-70: it is not clear what the point of this sentence is. What do the authors try to convey here and how does it relate to their experiments?

Response: We have amended the introduction to further clarify this section as follows:

Understanding how the somatosensory system responds to artificial forms of mechanical stimuli, and can distinguish between distinct forms of these stimuli, is important in interpreting studies that use these stimuli as measures of cortical excitability and plastic changes during disease or after injury. Whereas considerable literature exists for somatosensation within the healthy brain in non-human primates (for review, see [1,2]), literature on the somatosensory system of rodents is more limited. Given that rodents are the most prominent animal model used to study reaching and upper limb use and disability[3], it is important to understand how different forms of somatic and movement related stimuli used in rodent research are processed in the rodent somatosensory cortex. Most current research on the limb-associated somatosensory cortex in rodents utilize artificial forms of mechanical[4–7] or electrical stimuli[8,9] to assess sensory-evoked responses using electrophysiology of individual cells or aggregate responses from large cortical regions. Unfortunately, stimulus parameters used in these studies often vary from one study to the next, and few studies offer well defined rationale for their choice of stimuli, thereby making comparisons and interpretation across studies more challenging. Notably, artificial stimuli used for rodent research that may be thought of as relatively simple in comparison to naturalistic stimuli or movements, such as limb oscillation through a single axis of motion, still likely result in the generation of complex multimodal sensory information in the periphery. These multimodal signals may arise from activation of mechanoreceptors at the stimulator attachment site, the propagation of vibration at the fundamental and harmonic frequencies to distal Pacinian corpuscles closely associated with joints and bones[10], the activation of proprioceptive receptors within the muscles and joints, and to a large potential array of signals arising due to deflections of hair follicle associated mechanoreceptors (for review, see [11]). Although particular subsets of cutaneous and proprioceptive mechanoreceptors have long been known in non-human primates to be tuned to characteristics such as stimulus intensity, frequency and receptive field location[1,2,12–15], the harmonics resulting from square-wave oscillation of the entire forelimb or hindlimb of mice would likely elicit highly mixed activity in mechanoreceptor populations, with weighted preference to mechanoreceptor populations most sensitive to the particular frequencies and biomechanics of the stimulus.

Comment: - The authors use “limb movement” and “limb vibration” interchangeably throughout the manuscript. These are fundamentally different stimuli with each having its own specialized mechanoreceptors (Meissner and Pacinian corpuscles transduce vibrations and proprioceptors transduce limb movement).

Response: We believe the reviewer may have been referring to “limb movement” and “limb oscillation” that were used throughout the study. “Limb vibration” was not used to refer to our stimuli. We agree that “limb movement” could be interpreted as a unidirectional non-oscillitory movement of the limb that would be solely proprioceptive, and we have thus amended the manuscript to change all references of “limb movement” to “limb oscillation”. We have also changed all references of “vibrotactile” to “oscillatory”

Comment: - Page 6, line 123: can you give the precise coordinates of the imaging sites and how do these compare to those previously reported for the location of the forelimb and hindlimb S1?

Response: Imaging sites for the forelimb associated somatosensory region and the hindlimb associated somatosensory region were determined via flavoprotein autofluorescence intrinsic optical signal imaging as depicted in Fig 1a and as described in the methods sections “Identification of forelimb and hindlimb somatosensory cortex representations” and “Calcium imaging”.

Comment: - Page 6, line 132: why is the impedance of the pipette relevant for injections?

Response: This detail was included for replicability. The impedance of the pipette is proportional to the diameter of the opening at the tip of the pipette and determines the flow rate of solution out of the pipette into the tissue. Too quick of an injection holds the potential to damage tissue, so measures of the rate of OGB-1 indicator injection were closely monitored.

Comment: - Page 7, line 153: the authors should explain what “photon-shot noise” is.

Response: Photon-shot noise is a well characterized source of noise in a digitized image when using photodetectors to capture photons (see https://spie.org/samples/PM170.pdf). We have amended the text to increase clarity for this section.

“Using custom scripts written in Metamorph (Molecular Devices, California U.S.A.), a median filter (radius, 1 pixel) was applied to each of the image sequences of 8 trial sweeps at each frequency and duration of stimulation in order to remove photodetector related photon transfer noise (e.g. photon-shot noise) from the digitized images.”

Comment: - Page 8, line 169: a detailed description of the AUC analysis is missing. What exactly is being classified here? What does “half-width” refer to? Where is time-to-peak analyzed in the results?

Response: Time-to-peak did not vary by stimulus and was removed from the manuscript. Half-width is highly correlated with AUC and redundant, so it was also removed from the manuscript. We have amended the manuscript to further describe AUC as follows.

Methods

Neuronal traces that met these criteria were included in subsequent analysis of peak amplitude and area under the curve (AUC) measurements. AUC was measured as the total area under the curve of the ΔF/Fo from when the stimulus began to when the ΔF/Fo returned to the baseline level. AUC was used as a measurement of the sum strength of the calcium transient response over the time course of that particular transient.

Comment: - Page 8, lines 179-181: was the data z-scored prior to calculating the correlation coefficients?

Response: Yes, Pearson’s product-moment correlation coefficient is used to generate r values correlating the deltaF/Fo of neurons.

Comment: - Page 9, line 189: a better description of the “seed-based” analysis is needed.

Response: Manuscript has been amended with further description as follows:

In order to demonstrate a visual example of how the calcium fluorescence of a single cell may correlate with other pixels across an imaging window, a custom Matlab R2018a (Mathworks) script was used to generate an example seed-based stimulus-evoked neuronal correlation map (as shown in Fig 4a). With this script, filtered image sequences of ΔF/Fo were imported into Matlab and the signal trace for a chosen neuron was correlated with the ΔF/Fo of all pixels within the imaging field (a representative image during 1s 100Hz stimulus measured at a depth of 130μm in the cHL region is shown in Fig 4a). 

Comment: - The stimulus evoked trace in figure 2C does not appear to be an actual calcium transient, but looks more like noise fluctuations. Is this the best available example for a 10 Hz selective neuron? Is this bump that appears in the average trace only due to a single trial or does it reliably appear across multiple trials?

Response: We have amended Fig 2B,C with two better example neurons that demonstrated above threshold responses to all stimuli with preferential activation by high frequency stimuli in B, and strongest activation by the 10Hz stimulus in C.

Comment: - In general, the authors cannot rule out frequency selective responses because of the limited number of frequencies tested. We do not know what happens beyond 300 Hz.

Response: We agree with the reviewer that an extended frequency range and more comprehensive number of tested frequencies may demonstrate cells with responses selectively tuned to certain high frequencies as demonstrated in Prsa et al., 2019.

Comment: - Page 12, lines 266-267: could this be due to saturated Ca responses? In general, why is the decay of the calcium transient relevant? What does it reflect exactly?

Response: If we were to speculate, a longer decay tau for the population response to high frequency stimuli may indicate that the neuronal population takes longer to desensitize in its firing rate over the period of the 1 second stimulation leading to a longer calcium transient. This however would have to be shown in future studies measuring the desensitization of spiking activity using electrical neurophysiology instead of calcium imaging due to the slow temporal dynamics of calcium imaging.

Comment: - Page 14, lines 299-300: or could it be due to the fact that nearby neurons have more similar expression levels?

Response: DeltaF/Fo was calculated from the fluorescence measurements of each of the cells which normalizes for fluorescence level of the cell, and Pearson’s r was used for correlation analysis, so we support our initial interpretation.

Comment: - Page 14, lines 308-310: how does this compare to analysing the pre- or post- stimulus baseline period and also is it significant when running a randomization test with shuffled data?

Response: We have added an additional stationary bootstrapping procedure to create figure 4c,d that displays correlation values that were greater than the 97.5th percentile of the bootstrap. A 5 frame mean block length for the stationary bootstrap was used as this corresponds to a block length of 200ms for our 25Hz sampling rate, the minimum period of time between current pulses necessary to detect a calcium transient for independent action potentials measured using OGB-1 (Tada et al., 2014). The differences observed remain significant after the bootstrapping procedure.

Comment: - Page 14, lines 312-313: could this simply be due to the fact that higher frequencies have stronger and more frequent responses and thus yield higher correlation coefficients? Was Z scoring performed?

Response: Pearson’s r between the deltaF/Fo of each cell was used in the correlation analysis. We agree with the reviewers interpretation; it is likely that the stronger and more frequent responses to higher frequency stimuli directly lead to the higher average cross-correlation measured.

Comment: - Page 15, lines 340-342: It is not clear what this means or implies.

Response: We have amended the manuscript text as follows:

Thus, the low percent overlap for these 10 oscillation stimuli and the trend towards dissimilarity when comparing their correlation maps (Fig 5a-d) is a better indicator of differences between the population responses to these 10 oscillation stimuli than the averaged response characteristics of these populations (Fig5e-h).

Comment: - Page 16, lines 351-352: the authors compare 0.1 s and 0.05 s stimulus durations with 1 s durations (a 10 to 20 fold duration difference). Why such big differences and such short stimuli? A 50 ms vibration will hardly evoke any spikes.

Response: If somatosensory responses were primarily dependent on the frequency of the stimulus, and not modulated by the duration of the stimulus, then we’d expect both short and long duration stimuli to show similarities in their evoked somatosensory responses and/or the pattern of correlation within their cell-cell correlation maps. We show that this is not the case, and that the duration of the stimulus has a significant effect on the pattern of the cortical response.

Comment: - Page 16, lines 359-360: how do the authors explain this?

Response: Our study is not able to define why within trial comparisons for the 100Hz stimuli were significantly different from the between trial comparisons of 0.1s and 1s 100Hz stimuli, whereas this was not the case for the 200Hz stimuli. If we were to speculate, we may hypothesize that as the frequency of the stimulus increases past a certain frequency threshold for the particular mechanical stimulator used in this study there is a shift towards more weighted activation of Pacinian corpuscles and other high frequency sensitive receptors leading to more similar cortical representation between the two durations of 200Hz stimuli, but this would have to be shown with further research.

Comment: - The analysis of the population dF/F (Fig 3A) does not provide much insight. While not incorrect, there is little gain in understanding of encoding properties by showing the population as a whole is more active when the authors have shown in the previous figure that the number of neurons recruited is increased for higher frequency stimuli. The amplitude, decay time and AUC are all strongly related to each other, as calcium signals are subject to decay. It is unclear what the additional functional significance is of the AUC or decay time being different across stimulation frequencies.

Response: We agree with the reviewer that the analysis of population dF/F does not provide any information on encoding properties of a population responding to different stimuli. It does, however, demonstrate that the stronger cortical responses to high frequency stimuli are not well captured within the average response amplitude across the population, but are instead more significantly captured by the duration of the fluorescence increase (indicated by greater AUC and decay times) and the percent of neurons that show an increase in fluorescence indicative of more than one action potential underlying their calcium transient - what we have termed "strongly responding" neurons here (now described in methods).

---

## [Decision Letter · Decision Letter 1]

25 Jan 2021

PONE-D-20-21079R1

Distinct patterns of activity in individual cortical neurons and local networks in primary somatosensory cortex of mice evoked by square-wave mechanical limb stimulation

PLOS ONE

Dear Dr. Winship,

Thank you for submitting your manuscript to PLOS ONE. After careful consideration, we feel that it has merit but does not fully meet PLOS ONE’s publication criteria as it currently stands. Therefore, we invite you to submit a revised version of the manuscript that addresses the points raised during the review process.

Please, revise the manuscript address concerns raised by reviewer #1.

Abstract:

- Lines 36-39: Please, rephrase for clarity : “process dissimilar stimuli with distinct patterns of activity” and “represent distinct […] stimuli with unique patterns of activity”.

Introduction:

- Lines 58-59: Please, provide references to studies using square-wave pulses “as measures of cortical excitability and plastic changes during disease or after injury”?

- Lines 60-61: This sentence should be nuanced. Literature on the somatosensory system of rodents is quite extensive if taking into account the vibrissae system.

- Lines 61-62: Rodents are certainly not the “most prominent animal model used to study reaching and upper limb use and disability”. Sensorimotor studies on the rodent upper limb system have emerged only relatively recently. This system has been traditionally and most prominently studied in non-human primates. Also the reference that is cited here ([3]) has nothing to do with upper limb use and reaching, it instead provides a general comparison between rats and mice as models in biomedical research.

- Lines 67-73: Please, edit for clarity.

- Lines 96-98: Please, consider rewriting by more specifically stating the actual findings.

Methods

- Lines 142-143: Can the authors provide the coordinates relative to bregma of the identified cFL and cHL areas? This was asked in the previous revision but was not answered!

- In the initial review, I asked the authors to provide information about the actual amplitude of vibrations produced at each frequency. The authors replied that they amended the methods with the following information: “During stimulation, the entire limb underwent an oscillation with the following peak-peak amplitudes by frequency: 280um (3 & 10Hz), 335um (100Hz), 220um (200Hz), 170um (300Hz).”.

No detail is however given about how these amplitudes were measured! The authors mention that they “imaged the limb oscillations” with no additional technical detail. A 300 Hz oscillation would require a minimum 600 fps frame rate to capture the movement amplitude. Also, the following questions were not answered: At what sampling rate were the oscillations produced? How was the stimulus integrity assured? Was it consistent across several repetitions?

Results

Fig. 2B,C: The traces correspond to averages of 8 trials (no information about trail-to-trial variability is given). Can the authors also show the individual Ca traces from every trial? What do the polygons in the bottom panels represent (they are not mentioned in the figure legend)? Also, the labels on these polygons are tiny and thus unreadable.

Lines 268-270: This is entirely qualitative. Can the authors demonstrate statistically that the responses of these example neurons show stimulus specificity?

Fig.2: According to the activity maps in Fig.2A and 2D, a relatively large number of neurons is driven by the vibrotactile stimulation. This is in contradiction to what has previously been reported for L2/3 of the somatosensory cortex. Even in awake behaving mice, evoked activity by tactile stimulation remains very sparse. O’Connor et al. (Neuron, 2010) found that “A sparse subset of L2/3 neurons showed robust fluorescence transients […] high event rates were seen in only a small subset of neurons, with the majority showing low, near zero, event rates” in the vibrissae S1. Prsa et al. (Nature, 2019) reported 1,285 responsive neurons from 75 fields of view in the forelimb S1, yielding ca. 17 neurons per field of view. The authors found a much larger number of responding neurons under anaesthesia (Fig.2A,D). Can the authors quantify these numbers and how do they explain the discrepancy with previous studies?

The authors also avoided addressing the following comment from the previous revision:

“Why was such a high amplitude chosen? Previous studies in rodents used an order of magnitude smaller amplitudes (in the um range) which evoked highly reliable responses and is already well beyond their perceptual thresholds. It therefore seems highly plausible that the 1 mm oscillations evoked highly saturated neural responses, which are definitely not in the natural perceptual range of mice.”

The following question from the previous revision was not answered:

“A threshold criterion was used to identify responsive neurons. Was this analysis done on the average traces or on every single trial? The authors claim that this procedure was “optimal for separating responsive neurons from noise”, but based on what criteria?”

The authors still state in the manuscript that their criterion is “optimal”. How is optimality assessed? Compared to what?

The authors gave the following answer to the question regarding the analysis of the Ca transient decay:

“If we were to speculate, a longer decay tau for the population response to high frequency stimuli may indicate that the neuronal population takes longer to desensitize in its firing rate over the period of the 1 second stimulation leading to a longer calcium transient. This however would have to be shown in future studies measuring the desensitization of spiking activity using electrical neurophysiology instead of calcium imaging due to the slow temporal dynamics of calcium imaging.”

I agree with the authors that this is highly speculative and also still do not understand why it is relevant to analyze the decay. I would suggest removing this analysis from the manuscript.

The following question from the previous revision was not answered:

“Page 12, lines 266-267: could this be due to saturated Ca responses?”

The authors answered the following in the previous revision:

“We agree with the reviewers interpretation; it is likely that the stronger and more frequent responses to higher frequency stimuli directly lead to the higher average crosscorrelation measured.”

The authors should therefore clearly state this in their manuscript.

We look forward to receiving your revised manuscript.

Kind regards,

Gennady Cymbalyuk, Ph.D.

Academic Editor

PLOS ONE

Reviewers' comments:

Reviewer's Responses to Questions

**Comments to the Author**

1. If the authors have adequately addressed your comments raised in a previous round of review and you feel that this manuscript is now acceptable for publication, you may indicate that here to bypass the “Comments to the Author” section, enter your conflict of interest statement in the “Confidential to Editor” section, and submit your "Accept" recommendation.

Reviewer #1: (No Response)

Reviewer #2: All comments have been addressed

Reviewer #3: All comments have been addressed

2. Is the manuscript technically sound, and do the data support the conclusions?

Reviewer #1: Partly

Reviewer #2: Yes

Reviewer #3: (No Response)

3. Has the statistical analysis been performed appropriately and rigorously? 

Reviewer #1: No

Reviewer #2: Yes

Reviewer #3: (No Response)

4. Have the authors made all data underlying the findings in their manuscript fully available?

Reviewer #1: Yes

Reviewer #2: Yes

Reviewer #3: (No Response)

5. Is the manuscript presented in an intelligible fashion and written in standard English?

Reviewer #1: Yes

Reviewer #2: Yes

Reviewer #3: (No Response)

6. Review Comments to the Author

Reviewer #1: The authors have done an unsatisfactory job in addressing the reviewers’ concerns and comments. Many of the questions from the initial revision were not answered or not properly addressed. These are detailed here below.

Moreover, the authors now state that the main contribution of their work is that the “characterization of the distinct patterns of neural activity elicited by such stimuli is of importance as these forms of squarewave stimulation are used in multiple fields outside of the study of sensory coding, such as in the observation of cortical recovery from damage or disease.”

It however remains unclear what exact implication these results can have on studies of cortical recovery from disease or other studies using similar sensory stimulation. The authors must do a better job arguing this point, beyond simple hand-waving. Otherwise, I really do not see the point of this manuscript and its results.

Abstract:

- Lines 36-39: I have trouble understanding the difference between “process dissimilar stimuli with distinct patterns of activity” and “represent distinct […] stimuli with unique patterns of activity”.

Introduction:

- Lines 58-59: Can the authors provide references to studies using square-wave pulses “as measures of cortical excitability and plastic changes during disease or after injury”?

- Lines 60-61: This sentence should be nuanced. Literature on the somatosensory system of rodents is quite extensive if taking into account the vibrissae system.

- Lines 61-62: Rodents are certainly not the “most prominent animal model used to study reaching and upper limb use and disability”. Sensorimotor studies on the rodent upper limb system have emerged only relatively recently. This system has been traditionally and most prominently studied in non-human primates. Also the reference that is cited here ([3]) has nothing to do with upper limb use and reaching, it instead provides a general comparison between rats and mice as models in biomedical research.

- Lines 67-69: Why is this point relevant here? Do the authors work towards solving this issue in their study?

- Lines 70-73: I don’t understand how this sentence relates to the previous one.

- Lines 96-98: I find this description too vague. Consider rewriting by more specifically stating the actual findings.

Methods

- Lines 142-143: Can the authors provide the coordinates relative to bregma of the identified cFL and cHL areas? This was asked in the previous revision but was not answered!

- In the initial review, I asked the authors to provide information about the actual amplitude of vibrations produced at each frequency. The authors replied that they amended the methods with the following information: “During stimulation, the entire limb underwent an oscillation with the following peak-peak amplitudes by frequency: 280um (3 & 10Hz), 335um (100Hz), 220um (200Hz), 170um (300Hz).”.

No detail is however given about how these amplitudes were measured! The authors mention that they “imaged the limb oscillations” with no additional technical detail. A 300 Hz oscillation would require a minimum 600 fps frame rate to capture the movement amplitude. Also, the following questions were not answered: At what sampling rate were the oscillations produced? How was the stimulus integrity assured? Was it consistent across several repetitions?

Results

Fig. 2B,C: The traces correspond to averages of 8 trials (no information about trail-to-trial variability is given). Can the authors also show the individual Ca traces from every trial? What do the polygons in the bottom panels represent (they are not mentioned in the figure legend)? Also, the labels on these polygons are tiny and thus unreadable.

Lines 268-270: This is entirely qualitative. Can the authors demonstrate statistically that the responses of these example neurons show stimulus specificity?

Fig.2: According to the activity maps in Fig.2A and 2D, a relatively large number of neurons is driven by the vibrotactile stimulation. This is in contradiction to what has previously been reported for L2/3 of the somatosensory cortex. Even in awake behaving mice, evoked activity by tactile stimulation remains very sparse. O’Connor et al. (Neuron, 2010) found that “A sparse subset of L2/3 neurons showed robust fluorescence transients […] high event rates were seen in only a small subset of neurons, with the majority showing low, near zero, event rates” in the vibrissae S1. Prsa et al. (Nature, 2019) reported 1,285 responsive neurons from 75 fields of view in the forelimb S1, yielding ca. 17 neurons per field of view. The authors found a much larger number of responding neurons under anaesthesia (Fig.2A,D). Can the authors quantify these numbers and how do they explain the discrepancy with previous studies?

The authors also avoided addressing the following comment from the previous revision:

“Why was such a high amplitude chosen? Previous studies in rodents used an order of magnitude smaller amplitudes (in the um range) which evoked highly reliable responses and is already well beyond their perceptual thresholds. It therefore seems highly plausible that the 1 mm oscillations evoked highly saturated neural responses, which are definitely not in the natural perceptual range of mice.”

The following question from the previous revision was not answered:

“A threshold criterion was used to identify responsive neurons. Was this analysis done on the average traces or on every single trial? The authors claim that this procedure was “optimal for separating responsive neurons from noise”, but based on what criteria?”

The authors still state in the manuscript that their criterion is “optimal”. How is optimality assessed? Compared to what?

The authors gave the following answer to the question regarding the analysis of the Ca transient decay:

“If we were to speculate, a longer decay tau for the population response to high frequency stimuli may indicate that the neuronal population takes longer to desensitize in its firing rate over the period of the 1 second stimulation leading to a longer calcium transient. This however would have to be shown in future studies measuring the desensitization of spiking activity using electrical neurophysiology instead of calcium imaging due to the slow temporal dynamics of calcium imaging.”

I agree with the authors that this is highly speculative and also still do not understand why it is relevant to analyse the decay. I would suggest removing this analysis from the manuscript.

The following question from the previous revision was not answered:

“Page 12, lines 266-267: could this be due to saturated Ca responses?”

The authors answered the following in the previous revision:

“We agree with the reviewers interpretation; it is likely that the stronger and more frequent responses to higher frequency stimuli directly lead to the higher average crosscorrelation measured.”

The authors should therefore clearly state this in their manuscript.

Reviewer #2: After the revision quality of the manuscript was significantly improved and, in my opinion, it is ready for publication.

Reviewer #3: (No Response)

7. PLOS authors have the option to publish the peer review history of their article (what does this mean?). If published, this will include your full peer review and any attached files.

Reviewer #1: No

Reviewer #2: No

Reviewer #3: No

---

## [Author Response · Author response to Decision Letter 1]

11 Mar 2021

Reviewer #1: The authors have done an unsatisfactory job in addressing the reviewers’ concerns and comments. Many of the questions from the initial revision were not answered or not properly addressed. These are detailed here below.

Moreover, the authors now state that the main contribution of their work is that the “characterization of the distinct patterns of neural activity elicited by such stimuli is of importance as these forms of squarewave stimulation are used in multiple fields outside of the study of sensory coding, such as in the observation of cortical recovery from damage or disease.”

It however remains unclear what exact implication these results can have on studies of cortical recovery from disease or other studies using similar sensory stimulation. The authors must do a better job arguing this point, beyond simple hand-waving. Otherwise, I really do not see the point of this manuscript and its results.

We thank the reviewer for his thorough review of the manuscript and suggestions for amendments, and apologize if our previous revisions were not sufficient. We have amended the introduction to further highlight the implications of our study for understanding somatosensory representations of mechanical square-wave stimuli under anesthetized conditions as follows:

Investigations of how the somatosensory cortex responds to artificial forms of stimulation, and to what extent patterns of regional and cellular activity can distinguish between distinct stimuli with overlapping characteristics, is important in interpreting studies that use such stimuli to elicit cortical responses in the healthy brain, or as a measure of cortical excitability and plasticity during disease or after injury [1–5]. While square-wave and sinusoidal patterns of mechanical limb stimulation have been used in studies after cortical injury, and those studies identified deficits in the amplitude and fidelity of evoked activity after damage to the somatosensory cortex (for example see [1,2,4,6]), a more detailed investigation across multiple frequencies and intensities of stimulation has not been performed with mechanical square-wave stimuli. Mechanical square-wave stimulation allows precise control of the fundamental frequency (as with sinusoidal stimulation) but includes harmonic frequencies. These harmonics may overlap between different stimuli, and it may, therefore, be more challenging for the somatosensory system to distinguish between these stimuli. Whereas considerable literature exists for somatosensation within the healthy brain in non-human primates (for review, see [7,8]), and the sensory-evoked response properties of the barrel cortex of rodents has been extensively studied [9,10], literature on the sensory evoked response properties of limb associated somatosensory system of rodents is more limited.

Abstract:

- Lines 36-39: I have trouble understanding the difference between “process dissimilar stimuli with distinct patterns of activity” and “represent distinct […] stimuli with unique patterns of activity”.

We have amended the abstract as follows:

Artificial forms of mechanical limb stimulation are used within multiple fields of study to determine the level of cortical excitability and to map the trajectory of neuronal recovery from cortical damage or disease. Square-wave mechanical or electrical stimuli are often used in these studies, but a characterization of sensory-evoked response properties to square-waves with distinct fundamental frequencies but overlapping harmonics has not been performed. To distinguish between somatic stimuli, the primary somatosensory cortex must be able to represent distinct stimuli with unique patterns of activity, even if they have overlapping features. Thus, mechanical square-wave stimulation was used in conjunction with regional and cellular imaging to examine regional and cellular response properties evoked by different frequencies of stimulation. Flavoprotein autofluorescence imaging was used to map the somatosensory cortex of anaesthetized C57BL/6 mice, and in vivo two-photon Ca2+ imaging was used to define patterns of neuronal activation during mechanical square-wave stimulation of the contralateral forelimb or hindlimb at various frequencies (3, 10, 100, 200, and 300 Hz). The data revealed that neurons within the limb associated somatosensory cortex responding to various frequencies of square-wave stimuli exhibit stimulus-specific patterns of activity. Subsets of neurons were found to have sensory-evoked activity that is either primarily responsive to single stimulus frequencies or broadly responsive to multiple frequencies of limb stimulation. High frequency stimuli were shown to elicit more population activity, with a greater percentage of the population responding and greater percentage of cells with high amplitude responses. Stimulus-evoked cell-cell correlations within these neuronal networks varied as a function of frequency of stimulation, such that each stimulus elicited a distinct pattern that was more consistent across multiple trials of the same stimulus compared to trials at different frequencies of stimulation. The variation in cortical response to different square-wave stimuli can thus be represented by the population pattern of supra-threshold Ca2+ transients, the magnitude and temporal properties of the evoked activity, and the structure of the stimulus-evoked correlation between neurons.

Introduction:

- Lines 58-59: Can the authors provide references to studies using square-wave pulses “as measures of cortical excitability and plastic changes during disease or after injury”?

We have amended the introduction as follows:

Investigations of how the somatosensory cortex responds to artificial forms of stimulation, and to what extent patterns of regional and cellular activity can distinguish between distinct stimuli with overlapping characteristics, is important in interpreting studies that use such stimuli to elicit cortical responses in the healthy brain, or as a measure of cortical excitability and plasticity during disease or after injury [1–5]. While square-wave and sinusoidal patterns of mechanical limb stimulation have been used in studies after cortical injury, and those studies identified deficits in the amplitude and fidelity of evoked activity after damage to the somatosensory cortex (for example see [1,2,4,6]), a more detailed investigation across multiple frequencies and intensities of stimulation has not been performed with mechanical square-wave stimuli. Mechanical square-wave stimulation allows precise control of the fundamental frequency (as with sinusoidal stimulation) but includes harmonic frequencies. These harmonics may overlap between different stimuli, and it may, therefore, be more challenging for the somatosensory system to distinguish between these stimuli. Whereas considerable literature exists for somatosensation within the healthy brain in non-human primates (for review, see [6,7]), and the sensory-evoked response properties of the barrel cortex of rodents has been extensively studied [9,10], literature on the sensory evoked response properties of limb associated somatosensory system of rodents is more limited.

- Lines 60-61: This sentence should be nuanced. Literature on the somatosensory system of rodents is quite extensive if taking into account the vibrissae system.

We have amended the manuscript as follows to make it more nuanced:

Investigations of how the somatosensory cortex responds to artificial forms of stimulation, and to what extent patterns of regional and cellular activity can distinguish between distinct stimuli with overlapping characteristics, is important in interpreting studies that use such stimuli to elicit cortical responses in the healthy brain, or as a measure of cortical excitability and plasticity during disease or after injury [1–5]. While square-wave and sinusoidal patterns of mechanical limb stimulation have been used in studies after cortical injury, and those studies identified deficits in the amplitude and fidelity of evoked activity after damage to the somatosensory cortex (for example see [1,2,4,6]), a more detailed investigation across multiple frequencies and intensities of stimulation has not been performed with mechanical square-wave stimuli. Mechanical square-wave stimulation allows precise control of the fundamental frequency (as with sinusoidal stimulation) but includes harmonic frequencies. These harmonics may overlap between different stimuli, and it may, therefore, be more challenging for the somatosensory system to distinguish between these stimuli. Whereas considerable literature exists for somatosensation within the healthy brain in non-human primates (for review, see [6,7]), and the sensory-evoked response properties of the barrel cortex of rodents has been extensively studied [9,10], literature on the sensory evoked response properties of limb associated somatosensory system of rodents is more limited.

- Lines 61-62: Rodents are certainly not the “most prominent animal model used to study reaching and upper limb use and disability”. Sensorimotor studies on the rodent upper limb system have emerged only relatively recently. This system has been traditionally and most prominently studied in non-human primates. Also the reference that is cited here ([3]) has nothing to do with upper limb use and reaching, it instead provides a general comparison between rats and mice as models in biomedical research.

We agree with the reviewer on this point, and apologize for the error in citation. We have amended the introduction as follows:

Investigations of how the somatosensory cortex responds to artificial forms of stimulation, and to what extent patterns of regional and cellular activity can distinguish between distinct stimuli with overlapping characteristics, is important in interpreting studies that use such stimuli to elicit cortical responses in the healthy brain, or as a measure of cortical excitability and plasticity during disease or after injury [1–5]. While square-wave and sinusoidal patterns of mechanical limb stimulation have been used in studies after cortical injury, and those studies identified deficits in the amplitude and fidelity of evoked activity after damage to the somatosensory cortex (for example see [1,2,4,6]), a more detailed investigation across multiple frequencies and intensities of stimulation has not been performed with mechanical square-wave stimuli. Mechanical square-wave stimulation allows precise control of the fundamental frequency (as with sinusoidal stimulation) but includes harmonic frequencies. These harmonics may overlap between different stimuli, and it may, therefore, be more challenging for the somatosensory system to distinguish between these stimuli. Whereas considerable literature exists for somatosensation within the healthy brain in non-human primates (for review, see [6,7]), and the sensory-evoked response properties of the barrel cortex of rodents has been extensively studied [9,10], literature on the sensory evoked response properties of limb associated somatosensory system of rodents is more limited. Given that rodents are a prominent animal model used to study recovery of reaching and upper limb use after central nervous system damage or disability [8,9], it is important to understand how different forms of somatic and movement related stimuli used in rodent research are processed in the rodent somatosensory cortex. Most current research on the limb-associated somatosensory cortex in rodents utilize artificial forms of mechanical[1,10–12] or electrical stimuli[13,14] to assess sensory-evoked responses using electrophysiology of individual cells or aggregate responses from large cortical regions.

- Lines 67-69: Why is this point relevant here? Do the authors work towards solving this issue in their study?

We agree that these comments were not relevant to the proposed work, and have removed these comments in the revised manuscript.

- Lines 96-98: I find this description too vague. Consider rewriting by more specifically stating the actual findings.

We have amended the section of the introduction as follows:

Calcium imaging has been used as a method for studying activity within regional cortical networks[1,21–24]. Here, Ca2+ imaging was used to investigate how the limb-associated somatosensory cortex represents different frequencies of square-wave limb oscillation, and if differential patterns of activity in somatosensory neurons and local networks could be detected within the Ca2+ response. Flavoprotein autofluorescence imaging[25–29] was first used to identify limb-associated regions of cortical activation during mechanical forelimb or hindlimb stimulation delivered with piezoelectric actuators[1,30] (Fig 1a). In vivo two-photon Ca2+ imaging was used to optically record the response properties of individual neurons and local neuronal networks within these limb-associated somatosensory regions during multiple frequencies of contralateral forelimb (cFL) or contralateral hindlimb (cHL) oscillation (Fig 1b-d). Our data shows that the magnitude of the neuronal population response in somatosensory cortex is non-linearly related to the frequency of mechanical limb stimulation. High frequency 100Hz, 200Hz and 300Hz stimuli were found to elicit more responding neurons, a greater response strength, and higher average cross-correlation between neurons than lower frequency 3Hz and 10Hz stimuli. Whereas average population response magnitude within the low frequency and high frequency groups was not a strong differentiator of how populations represented stimuli within these groups, the pattern of responsive neurons within the local neuronal network and differences observed in the cross-correlation maps for populations responding to these stimuli clearly differentiated between square-wave stimuli within and between stimulus groups. 

Methods

- Lines 142-143: Can the authors provide the coordinates relative to bregma of the identified cFL and cHL areas? This was asked in the previous revision but was not answered!

Please note that a coordinate based system was not used to define the exact position of the cFL and cHL, as the stereotaxic coordinates of these regions can vary from strain to strain, from colony to colony within a particular strain, and even to some degree from mouse to mouse within a colony. An area on the skull surface with coordinates +2mm to -2mm anterior-posterior, and +1mm to +5mm lateral to bregma was thinned, then flavoprotein IOS imaging performed using a low magnification (2.5X) objective capturing the entire thinned area. Flavoprotein IOS denoted the precisely mapped cortical areas responsive to the cFL or cHL, respectively. A craniotomy of approximately 3x3mm was then centered over the approximate centre of mass of the cFL/cHL region. The cFL and cHL regions mapped via flavoprotein IOS were used for injection of the OGB-1 indicator. These maps were also used to determine the precise locations for Ca2+ imaging within these regions. Approximate center of mass coordinates of the cHL was 2mm lateral, -1mm posterior. Approximate coordinates for the cFL was +1mm anterior, +2.5mm lateral from bregma. We have further amended the methods with the following to further clarify:

A 4 x 4 mm region of the skull overlying the right hemisphere somatosensory region was thinned to 25-50% of original thickness using a high-speed dental drill (~1-5mm lateral, +2 to -2 mm posterior to bregma). This thinned region was covered with 1.3% low-melt agarose dissolved in artificial CSF (ACSF) at 37oC, then covered with a 3mm glass coverslip. Flavoprotein autofluorescence (FA) imaging was performed through this thin skull preparation before in vivo Ca2+ imaging to determine somatosensory limb regions on the cortex. These mapped regions were later used for OGB-1 indicator injection and for determination of imaging window coordinates for Ca2+ imaging following the FA mapping. 

- In the initial review, I asked the authors to provide information about the actual amplitude of vibrations produced at each frequency. The authors replied that they amended the methods with the following information: “During stimulation, the entire limb underwent an oscillation with the following peak-peak amplitudes by frequency: 280um (3 & 10Hz), 335um (100Hz), 220um (200Hz), 170um (300Hz).”.

No detail is however given about how these amplitudes were measured! The authors mention that they “imaged the limb oscillations” with no additional technical detail. A 300 Hz oscillation would require a minimum 600 fps frame rate to capture the movement amplitude. Also, the following questions were not answered: At what sampling rate were the oscillations produced? How was the stimulus integrity assured? Was it consistent across several repetitions?

The movement of the limb was imaged using a Dalsa Pantera 1M60 camera mounted on a Leica SP5 microscope using a 2.5X objective. The limb was attached to the stimulator and placed on a black background to increase contrast of the limb relative to the background. Three different imaging frame rates and resolutions were recorded for each frequency of limb stimulus based on the limits imposed by the camera hardware: 58fps at 1024x1024, 100fps at 512x512 and 157fps at 256x256. As all of these imaging frame rates were undersampling relative to the Nyquist criterion for the 300Hz stimuli, we employed several measures to get an index of the peak-to-peak amplitude of the limb oscillation as follows. For all imaging framerates, we recorded extended video (upwards of 5 minutes per recording), then used max intensity projections to determine the maximal bending amplitude of the stimulus at each stimulus frequency. To confirm this method, we compared the max projection measurements and frame-per-frame measurements for low frequency stimuli and found them to be equal. Peak-to-peak deviation measurements from all three imaging framerates at all frequencies were compared and found not to be significantly different. To determine if the peak-to-peak bending amplitude was consistent across multiple stimulus trials, we repeated the video recordings multiple times and separately analyzed each trial. Bending amplitudes were consistent across trials of the same stimulus frequency. Bending actuators were driven by an A-M Systems Model 2100 stimulator, an analog stimulator with 1 microsecond timing (250 ns jitter). These methods are not included in the methods section, but could be added if warranted in the reviewer’s opinion.

Results

Fig. 2B,C: The traces correspond to averages of 8 trials (no information about trail-to-trial variability is given). Can the authors also show the individual Ca traces from every trial? What do the polygons in the bottom panels represent (they are not mentioned in the figure legend)? Also, the labels on these polygons are tiny and thus unreadable.

Figures have been revised with individual traces from every trial. Plots are revised and figure legend edited to mention that the plots demonstrate the peak Ca2+ response amplitude of the cell to each of the stimuli.

Lines 268-270: This is entirely qualitative. Can the authors demonstrate statistically that the responses of these example neurons show stimulus specificity?

We agree with the reviewer that these neurons are not showing specificity, only variance in their Ca2+ response magnitude to the different stimuli and preference for certain stimuli over others. The plots below the Ca2+ traces have been edited to make them more visible. These show the quantified response magnitude of the above neuron to each of the stimuli. We have amended the text to read as follows:

Fig 2b,c depict representative neurons selected from the imaging plane shown in A that display different magnitudes in their Ca2+ response to the different stimuli. Notably, the neuron in Fig 2b is more responsive to high frequency 100, 200, and 300Hz stimuli, whereas the neuron in Fig 2c exhibits a mean response that is transient and most strongly activated by 10Hz stimulation.

Fig.2: According to the activity maps in Fig.2A and 2D, a relatively large number of neurons is driven by the vibrotactile stimulation. This is in contradiction to what has previously been reported for L2/3 of the somatosensory cortex. Even in awake behaving mice, evoked activity by tactile stimulation remains very sparse. O’Connor et al. (Neuron, 2010) found that “A sparse subset of L2/3 neurons showed robust fluorescence transients […] high event rates were seen in only a small subset of neurons, with the majority showing low, near zero, event rates” in the vibrissae S1. Prsa et al. (Nature, 2019) reported 1,285 responsive neurons from 75 fields of view in the forelimb S1, yielding ca. 17 neurons per field of view. The authors found a much larger number of responding neurons under anaesthesia (Fig.2A,D). Can the authors quantify these numbers and how do they explain the discrepancy with previous studies?

Our results demonstrated approximately 15% of cells exhibiting a significant response to 3Hz or 10Hz stimuli, and approximately 30% of the population responding for each of the 100, 200, or 300Hz stimuli. This is quantified in Fig. 2E. These results are consistent with Hayashi et al., 2018 that performed a similar set of experiments using a vibrotactile stimulus applied to the limb of the mouse, also under anesthetized conditions. Note that the activity maps in Fig. 2A display not only deltaF/Fo within neurons, but also within the neuropil surrounding neurons. We have amended the results with the following:

Notably, high frequency 100 & 200Hz stimuli displayed a significantly greater percentage of above threshold responses relative to 3 & 10Hz stimuli. Consistent with previous research by Hayashi et al., 2018, approximately 15% of neurons exhibited a significant response for the 3Hz or 10Hz stimuli, and approximately 30% of the population exhibited a significant response for each of the 100, 200, or 300Hz stimuli (Fig 2e). The majority of neurons with above threshold Ca2+ responses were responding to multiple stimuli, with approximately 32.7 ± 4.2% selective to a single stimulus, 33.2 ± 1.8% activated by two stimuli, and 34.1 ± 3.8% activated by three or more stimuli (Fig 2f).

The authors also avoided addressing the following comment from the previous revision:

“Why was such a high amplitude chosen? Previous studies in rodents used an order of magnitude smaller amplitudes (in the um range) which evoked highly reliable responses and is already well beyond their perceptual thresholds. It therefore seems highly plausible that the 1 mm oscillations evoked highly saturated neural responses, which are definitely not in the natural perceptual range of mice.”

and

The following question from the previous revision was not answered: “Page 12, lines 266-267: could this be due to saturated Ca responses?”

The stimulator moved the entire limb and was not directed to a focal location on the glabrous skin. The large amplitude stimuli were chosen to elicit strong activity in proprioceptive receptor populations of the limb in addition to cutaneous receptor activation at the site of stimulator attachment. We agree with the reviewer that the stimuli are not naturalistic, however it is speculation on whether the neural responses were saturated or not. We have amended the discussion with the following text to address this possibility:

This study did not attempt to directly examine the contribution of specific mechanoreceptor populations in the limbs to the afferent signal generated by our square-wave stimuli and their corresponding cortical responses. While it is not possible to give specific predictions on how strongly each modality was represented within the afferent response and cortical activity, peripheral mechanoreceptors are known to have varied activation thresholds at a range of frequencies[75,76], and the overall pattern of evoked activity would therefore reflect mixed afferent signals generated by the stimuli in this study. Saturation of the calcium response for high frequency limb oscillation could potentially contribute to similarities in the percent of neurons responding (Fig 2e) and the average population response magnitude (Fig 3) to our high frequency stimuli. However, distinct patterns of cellular responses for neurons responsive to each of the high frequency stimuli (Fig 2g), and differences observed in the correlational map structure between different high frequency stimuli (Figs 5,6), suggest that the calcium response maintained sufficient fidelity to record distinct representations of stimulus frequency. Future studies could make use of measurement methods with higher temporal fidelity and transgenic animals expressing channelrhodopsins in genetically-identified modality specific afferents to further examine the contribution of particular modalities to the cortical response[62].

The following question from the previous revision was not answered:

“A threshold criterion was used to identify responsive neurons. Was this analysis done on the average traces or on every single trial? The authors claim that this procedure was “optimal for separating responsive neurons from noise”, but based on what criteria?” The authors still state in the manuscript that their criterion is “optimal”. How is optimality assessed? Compared to what?

The threshold criterion was chosen as an effective method to identify signal from noise in averaged traces consistent with previous studies (e.g. Roth et al., 2012, Winship et al., 2007,2008). We have further amended the methods as follows to highlight how this was assessed:

Due to the large number of signal traces that required analysis (over 40,000 traces; over 6000 neurons; 10 mice), a range of threshold criteria, based on previous research [1,23,32–34], were tested to differentiate responsive neuron Ca2+ transients from fluorescent noise. This range of threshold criteria was tested against manual annotation of Ca2+ responses gathered from multiple observers for a small subset of the experimental dataset from multiple animals. Positive identification of a Ca2+ transient from noise within the manual annotation was based on the expected Ca2+ transient waveform demonstrating fast rise on the leading edge of the fluorescence, and a slow decay back to baseline. The threshold criteria found to most effectively select Ca2+ transient waveforms that met this expected waveform shape was selected for identification of Ca2+ transients across all animals. A threshold criteria requiring the Ca2+ fluorescence of the cell, averaged from 8 trials, to increase by 3X the standard deviation of the baseline period ΔF/Fo (baseline defined as 1s before stimulus onset), and remain above this criteria for 160ms (4 successive frames), was used to differentiate the Ca2+ transient waveform of a response of the neuron from random noise fluctuations.

The authors gave the following answer to the question regarding the analysis of the Ca transient decay:

“If we were to speculate, a longer decay tau for the population response to high frequency stimuli may indicate that the neuronal population takes longer to desensitize in its firing rate over the period of the 1 second stimulation leading to a longer calcium transient. This however would have to be shown in future studies measuring the desensitization of spiking activity using electrical neurophysiology instead of calcium imaging due to the slow temporal dynamics of calcium imaging.”

I agree with the authors that this is highly speculative and also still do not understand why it is relevant to analyse the decay. I would suggest removing this analysis from the manuscript.

We agree that this is speculative, and accordingly have removed the decay analysis from the manuscript as suggested.

The authors answered the following in the previous revision:

“We agree with the reviewers interpretation; it is likely that the stronger and more frequent responses to higher frequency stimuli directly lead to the higher average crosscorrelation measured.” The authors should therefore clearly state this in their manuscript.

We have amended the discussion with the following:

A greater average cross-correlation for high frequency stimuli was observed (Fig 4e), potentially due to stronger and more frequent responses within the neural population to these high frequency stimuli.

---

## [Decision Letter · Decision Letter 2]

7 Apr 2021

PONE-D-20-21079R2

Distinct patterns of activity in individual cortical neurons and local networks in primary somatosensory cortex of mice evoked by square-wave mechanical limb stimulation

PLOS ONE

Dear Dr. Winship,

Thank you for submitting your manuscript to PLOS ONE. After careful consideration, we feel that it has merit but does not fully meet PLOS ONE’s publication criteria as it currently stands. Therefore, we invite you to submit a revised version of the manuscript that addresses the points raised during the review process.

-Please, provide description of how limb movement amplitudes were measured in the Methods section and

consider adding reference 23 to the sentence: " ... and transgenic animals expressing channelrhodopsins in genetically-identified modality specific afferents to further examine the contribution of particular modalities to the cortical response[62]. "

We look forward to receiving your revised manuscript.

Kind regards,

Gennady S. Cymbalyuk, Ph.D.

Academic Editor

PLOS ONE

Journal Requirements:

Reviewers' comments:

Reviewer's Responses to Questions

**Comments to the Author**

1. If the authors have adequately addressed your comments raised in a previous round of review and you feel that this manuscript is now acceptable for publication, you may indicate that here to bypass the “Comments to the Author” section, enter your conflict of interest statement in the “Confidential to Editor” section, and submit your "Accept" recommendation.

Reviewer #1: All comments have been addressed

2. Is the manuscript technically sound, and do the data support the conclusions?

Reviewer #1: Yes

3. Has the statistical analysis been performed appropriately and rigorously? 

Reviewer #1: Yes

4. Have the authors made all data underlying the findings in their manuscript fully available?

Reviewer #1: Yes

5. Is the manuscript presented in an intelligible fashion and written in standard English?

Reviewer #1: Yes

6. Review Comments to the Author

Reviewer #1: Two final comments:

- I would suggest including the description of how limb movement amplitudes were measured in the Methods section.

- I suggest adding reference 23 to the sentence: " ... and transgenic animals expressing channelrhodopsins in genetically-identified modality specific afferents to further examine the contribution of particular modalities to the cortical response[62]. "

7. PLOS authors have the option to publish the peer review history of their article (what does this mean?). If published, this will include your full peer review and any attached files.

Reviewer #1: No

---

## [Author Response · Author response to Decision Letter 2]

13 Apr 2021

Reviewer #1: Two final comments:

- I would suggest including the description of how limb movement amplitudes were measured in the Methods section.

- I suggest adding reference 23 to the sentence: " ... and transgenic animals expressing channelrhodopsins in genetically-identified modality specific afferents to further examine the contribution of particular modalities to the cortical response[62]. "

We thank the reviewer for their final comments for our manuscript. We have added reference 23 to the sentence and have added the following description of how limb movement amplitudes were measured to the methods section:

During stimulation, the entire limb underwent an oscillation with the following peak-peak amplitudes by frequency based on the limb weight loaded electromechanical properties of the bending actuator: 280um (3 & 10Hz), 335um (100Hz), 220um (200Hz), 170um (300Hz). To determine these peak-peak oscillation amplitudes, the movement of the limb was imaged using a Dalsa Pantera 1M60 camera mounted on a Leica SP5 microscope using a 2.5X objective. The limb was attached to the stimulator and placed on a black background to increase contrast of the limb relative to the background. Bending actuators were driven by an A-M Systems Model 2100 stimulator, an analog stimulator with 1 microsecond timing (250 ns jitter). Three different imaging frame rates and resolutions were recorded for each frequency of limb stimulus based on the limits imposed by the camera hardware: 58fps at 1024x1024 (17.24ms exposure), 100fps at 512x512 (10ms exposure) and 157fps at 256x256 (6.37ms exposure). As all of these imaging frame rates were undersampling relative to the Nyquist criterion for the 300Hz stimuli, we employed several measures to get an index of the peak-to-peak amplitude of the limb oscillation as follows. For all imaging framerates, we recorded extended video (approximately 5 minutes per recording), then used max intensity projections to determine the maximal bending amplitude of the stimulus at each stimulus frequency. To confirm this method, we compared the max projection measurements and frame-per-frame measurements for low frequency stimuli and found them to be equal. Peak-to-peak deviation measurements from all three imaging framerates at all frequencies were compared and found not to be significantly different. To determine if the peak-to-peak bending amplitude was consistent across multiple stimulus trials, we repeated the video recordings multiple times and separately analyzed each trial. Bending amplitudes were consistent across trials of the same stimulus frequency.

---

## [Editor Report · Decision Letter 3]

16 Apr 2021

Distinct patterns of activity in individual cortical neurons and local networks in primary somatosensory cortex of mice evoked by square-wave mechanical limb stimulation

PONE-D-20-21079R3

Dear Dr. Winship,

We’re pleased to inform you that your manuscript has been judged scientifically suitable for publication and will be formally accepted for publication once it meets all outstanding technical requirements.

Kind regards,

Gennady S. Cymbalyuk, Ph.D.

Academic Editor

PLOS ONE
---

## [Editor Report · Acceptance letter]

20 Apr 2021

PONE-D-20-21079R3 

Distinct patterns of activity in individual cortical neurons and local networks in primary somatosensory cortex of mice evoked by square-wave mechanical limb stimulation 

Dear Dr. Winship:

I'm pleased to inform you that your manuscript has been deemed suitable for publication in PLOS ONE. Congratulations! Your manuscript is now with our production department. 

Kind regards, 

on behalf of

Dr. Gennady S. Cymbalyuk 

Academic Editor

PLOS ONE